# Accurate prediction of protein folding mechanisms by simple structure-based statistical mechanical models

Koji Ooka [1,2] & Munehito Arai [1,2,3] ✉

Recent breakthroughs in highly accurate protein structure prediction using deep neural networks have made considerable progress in solving the structure prediction component of the 'protein folding problem'. However, predicting detailed mechanisms of how proteins fold into specific native structures remains challenging, especially for multidomain proteins constituting most of the proteomes. Here, we develop a simple structure-based statistical mechanical model that introduces nonlocal interactions driving the folding of multidomain proteins. Our model successfully predicts protein folding processes consistent with experiments, without the limitations of protein size and shape. Furthermore, slight modifications of the model allow prediction of disulfide-oxidative and disulfide-intact protein folding. These predictions depict details of the folding processes beyond reproducing experimental results and provide a rationale for the folding mechanisms. Thus, our physics-based models enable accurate prediction of protein folding mechanisms with low computational complexity, paving the way for solving the folding process component of the 'protein folding problem'.

Recently, we have witnessed a remarkable leap in the prediction of three-dimensional protein structure from amino acid sequences by deep neural networks[1,2], and these breakthroughs have made significant progress toward solving the structure prediction component of the 'protein folding problem'[3]. Nevertheless, state-of-the-art protein structure prediction methods do not provide an understanding of how proteins fold into specific structures, i.e., the folding process component of the 'protein folding problem'[3,4]. Therefore, computational prediction of protein folding mechanisms based on protein structures readily available from experiments or machine learning remains a challenge. Many experimental studies have characterized the folding mechanisms of both single-domain proteins, which typically show simple two-state folding behavior[5], and multidomain proteins, which have many nonlocal interactions and exhibit more complicated folding behavior[6–9]. The experimentally observed folding pathways of small single-domain proteins have been successfully predicted

using an Ising-like simple statistical mechanical model, called the Wako−Saitô−Muñoz−Eaton (WSME) model, which is a coarse-grained structure-based model for proteins[10–24]. This model assumes that folding is initiated by local interactions between adjacent residues and propagates to distal regions via the growth and docking of native segments. Remarkably, the WSME model can be used to calculate the free energy landscapes of proteins, providing a comprehensive understanding of folding mechanisms, including folding pathways, kinetics, and the structures of intermediates and transition states. However, this model cannot be used for the prediction of the folding mechanisms of multidomain proteins. Furthermore, although long-time molecular dynamics calculations allow simulating the folding reactions of small single-domain proteins in up to 1 ms[25], they cannot simulate multidomain protein folding, which typically takes more than 100 ms[6–9]. As multidomain proteins are naturally abundant and constitute most of the proteomes[26], predicting the folding mechanisms

[1]Department of Physics, Graduate School of Science, The University of Tokyo, 3-8-1 Komaba, Meguro, Tokyo 153-8902, Japan. [2]Komaba Organization for Educational Excellence, College of Arts and Sciences, The University of Tokyo, 3-8-1 Komaba, Meguro, Tokyo 153-8902, Japan. [3]Department of Life Sciences, Graduate School of Arts and Sciences, The University of Tokyo, 3-8-1 Komaba, Meguro, Tokyo 153-8902, Japan. ✉e-mail: arai@bio.c.u-tokyo.ac.jp

of multidomain proteins from native structures is a key problem to be solved.

The folding reactions of multidomain proteins often involve multiple folding pathways and molten globule-like compact intermediates[6–9]. The intermediates accumulate via a hydrophobic collapse mechanism driven by nonlocal interactions between distant residues[27]. However, the WSME model does not allow for nonlocal interactions between distant residues in an amino acid sequence unless all intervening residues are folded (Fig. 1a, b, d). This precludes the folding of a discontinuous domain, consisting of residues separated in a sequence, prior to the folding of an intervening continuous domain, resulting in a failure to predict the folding mechanisms of multidomain proteins. In addition, the WSME model cannot explicitly consider the folding reactions of disulfide-intact proteins or those involving the oxidative formation of disulfide bonds[28], which are typical nonlocal interactions that stabilize many extracellular proteins[29].

To overcome these limitations, we developed a simple structure-based statistical mechanical model, named the WSME-L model (L denotes linker), which can introduce virtual linkers corresponding to nonlocal interactions anywhere in a protein molecule (Fig. 1c, e). This model improves the prediction of the folding mechanisms of small single-domain proteins compared with the original models. Surprisingly, our model successfully predicts the free energy landscapes that reproduce the experimentally observed folding behaviors of multidomain proteins. In addition, the models with slight modifications, named the WSME-L(SS) and WSME-L(SS$_{intact}$) models, allow the prediction of the folding processes involving oxidative disulfide bond formation and those of disulfide-intact proteins, respectively (Fig. 1f, g). These results suggest that the WSME-L models may pave the way for predicting protein folding mechanisms from native structures without the limitations of size and shape, and will be a useful tool for protein folding prediction in the post-AlphaFold era.

## Results

### WSME-L model

In the original WSME model, an Ising-like two-state variable, $m_k$, was assigned to each residue of the protein ($m_k = 1$ for native and 0 for other conformations). The Hamiltonian of the model is defined as:

$$H(\{m\}) = \sum_{i=1}^{N-1} \sum_{j=i+1}^{N} \varepsilon_{i,j} m_{i,j} \qquad (1)$$

where $N$ is the number of residues, and $\varepsilon_{i,j}$ is the contact energy between residues $i$ and $j$ in the native state[10–13]. The protein state $\{m\}$ represents a set of all residue states, $\{m_1, m_2, \cdots, m_N\}$, with $2^N$ possible conformations, and:

$$m_{i,j} = m_i m_{i+1} \cdots m_j = \prod_{k=i}^{j} m_k \qquad (2)$$

The free energy landscape can be calculated based on the exact analytical solution of the partition function[18]. An order parameter,

$$n = \frac{1}{N} \sum_{i=1}^{N} m_i \qquad (3)$$

which indicates the degree of native structure formation, was used as the reaction coordinate in the free energy landscape. A partition function restricted by the order parameter $n$ is denoted:

$$Z(n) = \mathrm{Tr}_n \exp\left[ -\frac{1}{k_B T} \left( H(\{m\}) - T \sum_{i=1}^{N} S_i m_i \right) \right] \qquad (4)$$

where $k_B$ is the Boltzmann constant, $T$ is the temperature, and $S_i$ (<0) is the entropic reduction of residue $i$ attributed to the formation of the native conformation. $\mathrm{Tr}_n$ is the sum of all possible states constrained by the value of the order parameter $n$. Two important assumptions were made in the WSME model. One assumption is that a protein is stabilized only by contacts present in the native structure. Another assumption is that native interactions between residues $i$ and $j$ are established only when all intervening residues fold cooperatively into their native conformations, that is, $m_{i,j} = 1$ (Fig. 1a, b).

To consider the nonlocal interactions between the N- and C-termini of a protein, Inanami et al. introduced a virtual linker at both termini[23]. Inspired by this, we developed a method to introduce a virtual linker between arbitrary residues $u$ and $v$ ($u < v$) and established a new model (WSME-L) that can consider all nonlocal interactions in a protein molecule. In this model, to represent nonlocal interactions through a virtual linker, we define $m_{i,j}^{(u,v)}$ as follows:

$$m_{i,j}^{(u,v)} = \left( \prod_{\min(i,u) \le k \le \max(u,i)} m_k \right) \left( \prod_{\min(v,j) \le k \le \max(j,v)} m_k \right) \qquad (5)$$

where residues $i$ and $j$ interact via the linker, if the following two consecutive regions are in their native conformations: (1) from residue $i$ to $u$ (or from residue $u$ to $i$) and (2) from residue $v$ to $j$ (or from residue $j$ to $v$) (Fig. 1c). Thus, residues $i$ and $j$ are consecutively connected as a "native stretch" through a linker shortcut, bypassing a long stretch of the main chain. The Hamiltonian of the WSME-L model with a single linker is expressed as follows:

$$H^{(u,v)}(\{m\}) = \sum_{i=1}^{N-1} \sum_{j=i+1}^{N} \varepsilon_{i,j} \left\lceil \frac{m_{i,j} + m_{i,j}^{(u,v)}}{2} \right\rceil \qquad (6)$$

where $\lceil \ \rceil$ is the ceiling function that prevents double counting. Thus, native contacts can be formed between the residues $i$ and $j$ if they are connected in a native stretch, either through the main chain ($m_{i,j}$) or through a linker ($m_{i,j}^{(u,v)}$).

To incorporate all interactions present in the native state of a protein, we defined the partition function of the WSME-L model as an ensemble of partition functions with a virtual linker introduced at each inter-residue contact, as follows:

$$Z_L(n) = Z(n) + \sum_{(u,v):\text{All contacts}} \left( Z^{(u,v)}(n) - Z(n) \right) \exp\left( \frac{S'^{(u,v)}(n)}{k_B} \right) \qquad (7)$$

where $Z^{(u,v)}(n)$ is a partition function with a virtual linker between residues $u$ and $v$, and $S'^{(u,v)}(n)$ is the entropy penalty corresponding to the reduced number of states owing to the virtual linker formation between residues $u$ and $v$ (see Methods for details). This WSME-L model corresponds to a generalization of the virtual linker model by Inanami et al.[23] and can consider all contacts present in the protein. We obtained an exact analytical solution for the partition function of the WSME-L model using the transfer matrix method[18] (see Methods for details). This solution allows us to calculate rigorous free energy landscapes with greatly reduced computational complexity compared to the direct calculation using Eq. (7).

### Folding of small single-domain proteins

To test whether the WSME-L model (Fig. 1e) can predict the folding mechanisms of small single-domain proteins with different topologies, we calculated the free energy landscapes of six small proteins for which the folding mechanisms have been extensively studied experimentally:

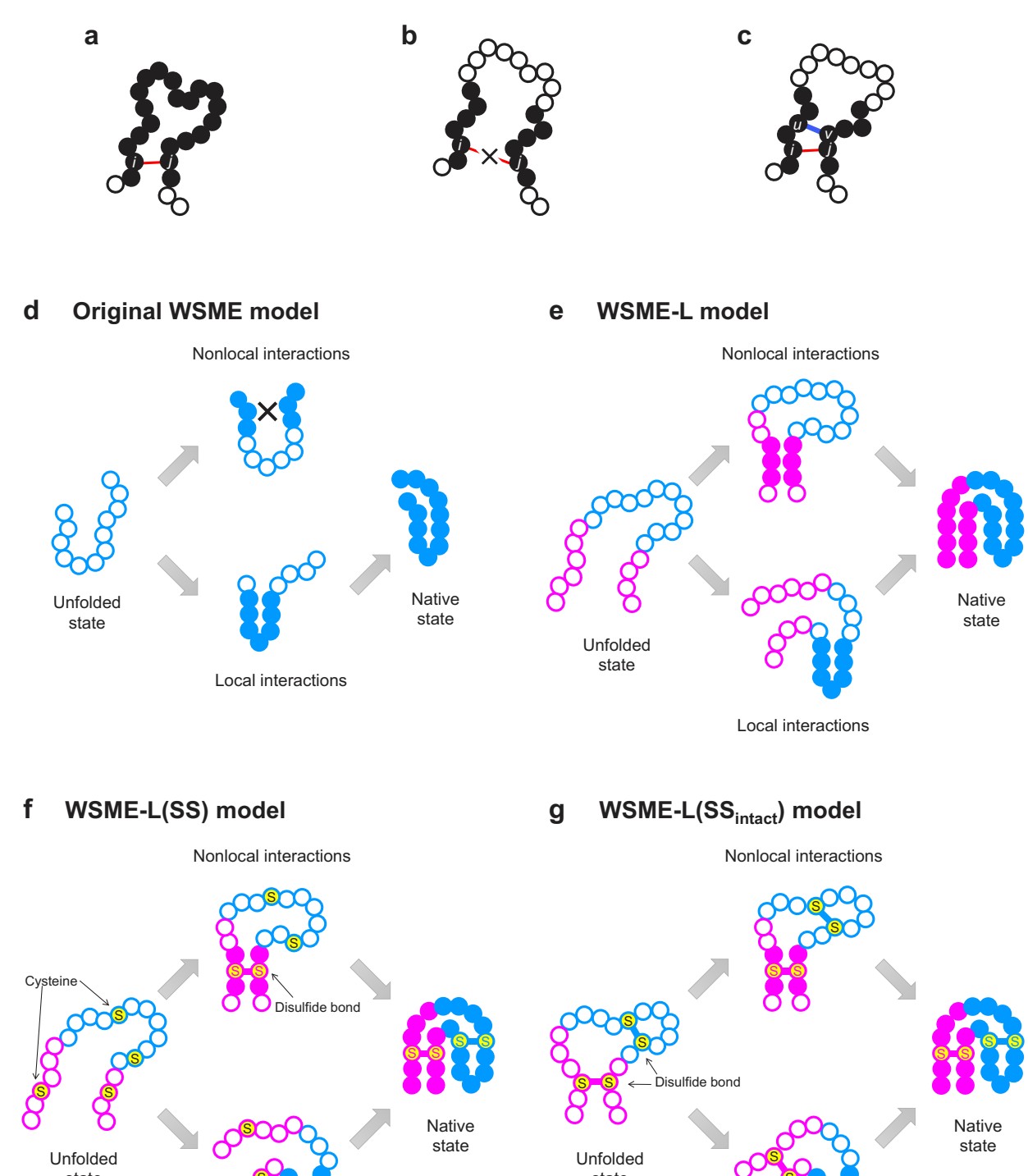

**Fig. 1 | Schematic representation of WSME and WSME-L models.** Residues in native or unfolded conformation ($m_k = 1$ or $0$) are shown as filled or open circles, respectively. **a**–**c** Assumption on native contact formation. In the original WSME model (**a**), native contact between residues $i$ and $j$ (red line) is formed only when these residues are connected by a native stretch along main chain; that is, when all intervening residues between them are in native conformation. A nonlocal native contact cannot be formed if two native stretches are not continuously connected along main chain (**b**). In the WSME-L model (**c**), nonlocal native contact between residues $i$ and $j$ separated in a sequence can be formed if a native stretch between

them is continuously connected through a linker shortcut at residues $u$ and $v$ (blue line), bypassing a long stretch of the main chain. **d** Original WSME model. Contact energy between residues $i$ and $j$, $\varepsilon_{i,j}$, can be uniform for all residues (Original model 1) or residue-dependent (Original model 2). **e** WSME-L model for the folding of small and large proteins without disulfide bonds. **f** WSME-L(SS) model for the folding involving oxidative disulfide bond formation. **g** WSME-L(SS$_{intact}$) model for the folding of disulfide-intact proteins. In (**e**–**g**), N- and C-terminal regions are shown by magenta circles and the intervening region is shown by blue circles. In (**f**, **g**), S denotes a cysteine residue.

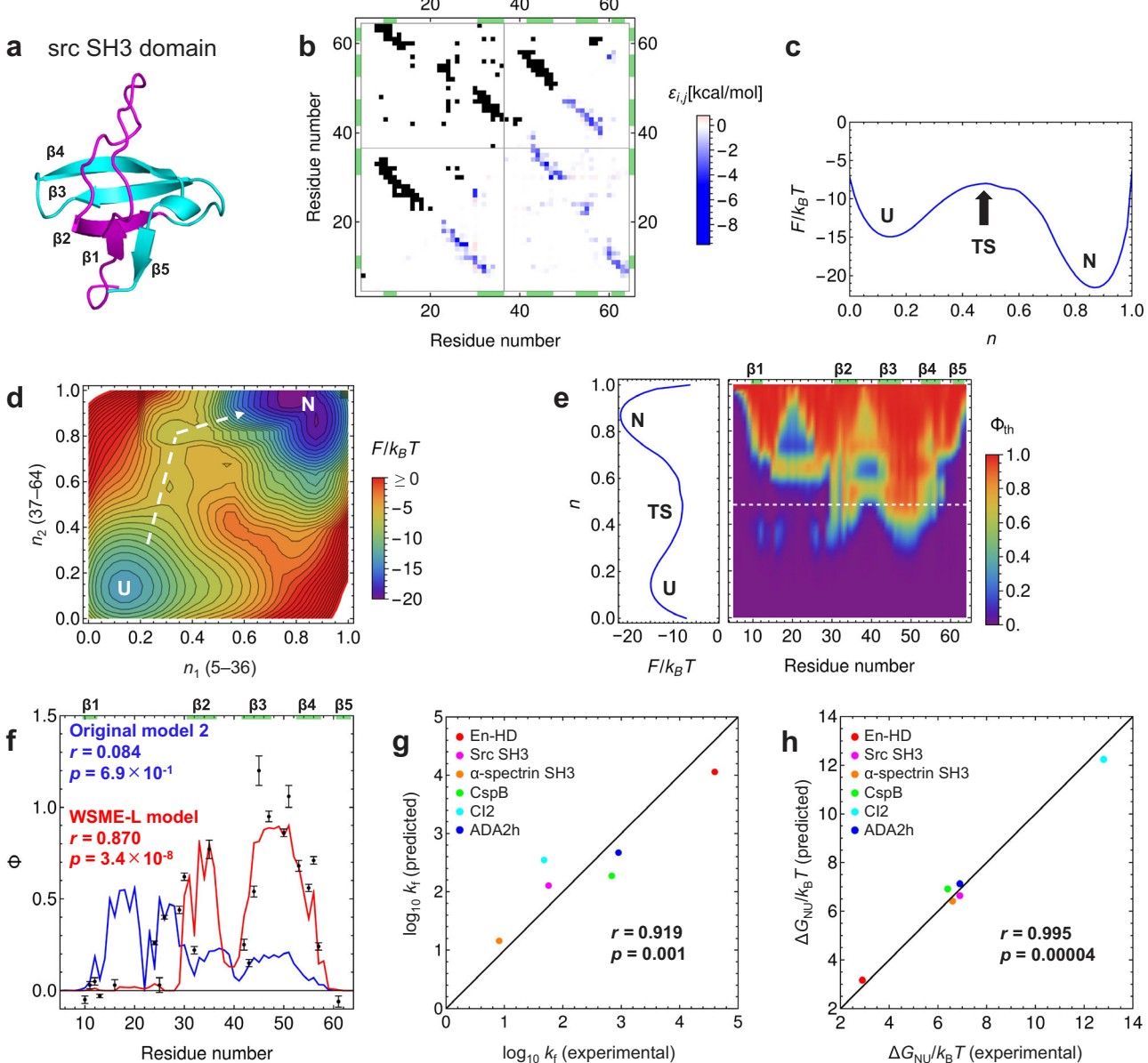

**Fig. 2 | Folding of small single-domain proteins predicted by WSME-L model.**
**a** Native structure of src SH3 domain. N-terminal (residues 5–36) and C-terminal (residues 37–64) halves are shown in magenta and cyan, respectively. **b** Contact map (top left) and AMBER-derived contact energy (bottom right) of src SH3 domain. **c** One-dimensional free energy landscape of src SH3 domain. U, TS, and N denote the unfolded, transition, and native states, respectively. **d** Two-dimensional free energy landscape of src SH3 domain. $n_1$ and $n_2$ are order parameters for N-terminal (residues 5–36) and C-terminal (residues 37–64) halves, respectively. Dominant folding pathway is indicated by white dashed line. Width and height of the panel are set proportional to the number of residues involved in regions for $n_1$

and $n_2$, respectively. **e** Residue-specific structure formation of src SH3 domain by theoretical Φ-value analysis. White dashed line indicates position of transition state. **f** Theoretical Φ-values predicted by Original model 2 (blue line) and WSME-L model (red line), and experimentally observed Φ-values (black filled circles) at transition state of src SH3 domain folding[33]. **g, h** Correlation between predicted and experimentally observed values of folding rate (**g**) and stability (**h**). In (**b**, **e**, **f**), green boxes on frame indicate locations of strands. In (**f**–**h**), correlation coefficients, $r$, and $p$-values of two-sided $t$-test without adjustments are shown. Source data are provided as a Source data file.

engrailed homeodomain (En-HD; SCOP2 classification[30]: all α), src SH3 domain (all β; Fig. 2a), α-spectrin SH3 domain (all β), cold shock protein B (CspB; all β), chymotrypsin inhibitor 2 (CI2; α + β), and activation domain of human procarboxypeptidase A2 (ADA2h; α + β) (Supplementary Table 1; see Methods for detailed calculations). The predictions of the WSME-L model were compared with those of two types of original WSME models: Original model 1 used a uniform contact energy, while Original model 2 used a residue-dependent weighted contact energy similar to that used in the WSME-L model.

The one-dimensional (1D) free energy landscapes predicted by these models indicated two-state folding reactions from the

unfolded to the native state for these proteins, in agreement with experimental results[31–36] (Supplementary Figs. 1–6 for all proteins; the results for the src SH3 domain are also shown in Fig. 2 as a representative). However, the rate constant for the folding reaction $k_f$ and the stability of the folded protein $\Delta G_{NU}$ predicted by the WSME-L model showed higher correlations with the experimental results than those obtained by the original models (Fig. 2g, h and Supplementary Fig. 7). We also calculated the two-dimensional (2D) free energy landscapes by defining the order parameters $n_1$ and $n_2$, corresponding to the degree of structure formation in the N- and C-terminal halves of each protein,

respectively, as follows:

$$\begin{cases} n_1 = \frac{1}{N_1} \sum\limits_{m_i \in \{m_1\}} m_i \\ n_2 = \frac{1}{N_2} \sum\limits_{m_i \in \{m_2\}} m_i \end{cases} \qquad (8)$$

where $\{m_1\}$ and $\{m_2\}$ are sets of residue states, and $N_1$ and $N_2$ are the number of residues in each region. The 2D free energy landscapes indicated that the structured region of each protein in the transition state was mainly localized in the N- or C-terminal half, as suggested by the experiments[31–36] (Fig. 2d and Supplementary Figs. 1–6).

To predict the detailed folding process at the amino acid residue resolution, we performed a theoretical Φ-value analysis (see Methods for details). Φ-value analysis has been widely used to experimentally study residue-specific structure formation during folding[31–36]. Φ = 0 or 1 when the residue is unfolded or fully folded, respectively. The theoretical Φ-values in the folding transition state calculated by the WSME-L model reproduced the experimentally observed Φ-values for all proteins well, and their correlation coefficients were higher than those of the original models (Fig. 2e, f and Supplementary Figs. 1–6). Interestingly, the WSME-L model can explain the differences in the folding pathways of two proteins with similar structures, src and α-spectrin SH3 domains (Supplementary Figs. 2 and 3). Therefore, the WSME-L model quantitatively improves the prediction accuracy of the folding mechanisms of small proteins compared with the original WSME models.

## Folding of large multidomain proteins

Next, to test whether the WSME-L model (Fig. 1e) can predict the folding mechanisms of large multidomain proteins with more than 100 residues, we applied the model to six well-studied proteins: apo-myoglobin (apoMb; all α; Fig. 3a), barnase (α + β; Fig. 3d), ribonuclease HI (RNase HI; α/β; Fig. 3g), dihydrofolate reductase (DHFR; α/β; Fig. 3j), α-subunit of tryptophan synthase (αTS; α/β; Fig. 3m), and indole-3-glycerol phosphate synthase (IGPS; α/β; Fig. 3p) (Supplementary Table 1; see Methods for detailed calculations). Kinetic folding experiments showed that apoMb accumulates an early folding intermediate in which the A, B, G, and H helices are formed among its eight helices (A–H). This then folds to the native state after forming the C, D, and E helices (Supplementary Fig. 8a)[37]. We calculated the 2D free energy landscape of apoMb using the original and WSME-L models with order parameters $n_1$ and $n_2$, which correspond to the degree of structure formation in the A, B, G, and H helices and the C, D, and E helices, respectively. The original models could not predict a free energy landscape consistent with the experimental results (Supplementary Figs. 9 and 10), because they assumed that the continuous region (C, D, and E helices) must fold before the discontinuous region (consisting of A and B helices on the N-terminal side and G and H helices on the C-terminal side). In contrast, the WSME-L model predicted the accumulation of a folding intermediate, in which the A, B, G, and H helices were formed, in the dominant folding pathway of apoMb (Fig. 3b). These results agree with the experimental results and are further supported by theoretical Φ-value analysis along the folding pathway (Fig. 3c).

For the folding reactions of barnase and RNase HI, the predictions of Original model 1 did not agree with the experimental results[38,39] (Supplementary Figs. 8b, c and 11–14). In contrast, the predictions of Original model 2 and the WSME-L model, both of which use residue-dependent weighted contact energies, agreed with the experimental results (Fig. 3d–i and Supplementary Figs. 8b, c and 11–14). However, the WSME-L model better predicted the Φ-values compared with Original model 2 (Supplementary Figs. 11 and 13).

DHFR consists of two domains, with one domain (the adenosine-binding domain, ABD) inserted into the other (the discontinuous loop domain, DLD) (Fig. 3j). Experiments have shown that the ABD of DHFR folds first, followed by DLD, and finally, both dock to form the native state via multiple folding pathways (Supplementary Fig. 8d)[40]. Although the original models did not explain this reaction (Supplementary Figs. 15–16), the extended WSME model with a single virtual linker between the N- and C-termini partially explained the experimental results[23]. However, the predicted intermediates were unstable, and multiple folding pathways were not predicted. In contrast, our WSME-L model reproduced the accumulation of stable intermediates, in which only ABD was formed, and the presence of multiple folding pathways (Fig. 3k, l and Supplementary Figs. 15 and 16).

αTS (268 residues) and IGPS (223 residues) are among the largest proteins whose folding mechanisms have been studied in detail by experiment[41,42]. They have similar native structures and show parallel folding pathways involving on-pathway intermediates, but their detailed structures differ between αTS and IGPS (Supplementary Fig. 8e, f). For αTS, all three models predicted the presence of two parallel pathways (Fig. 3n and Supplementary Figs. 17 and 18). However, only the WSME-L model was able to predict the formation of both N- and C-terminal regions in the on-pathway intermediate (Fig. 3o and Supplementary Figs. 17 and 18). Remarkably, although the original models failed to reproduce the experimentally observed folding processes of IGPS (Supplementary Figs. 19 and 20), the WSME-L model successfully reproduced the parallel pathways in IGPS folding (Fig. 3q) and accurately predicted the structure of the on-pathway intermediate in which strands 2–6 and helices 2–5 are formed (Fig. 3r). Again, the WSME-L model can distinguish subtle differences in folding mechanisms even for large proteins with similar structures.

In summary, the WSME-L model, which considers all nonlocal interactions, can predict the folding free energy landscapes of small to large proteins of different shapes in a unified manner.

## Folding with oxidative disulfide bond formation

Disulfide bonds are typical nonlocal interactions that stabilize a protein covalently and are present in many extracellular proteins or domains secreted outside cells[29]. Folding of such proteins involves oxidative disulfide bond formation. Oxidative folding reactions of RNase A and other proteins have long been studied experimentally, leading to Anfinsen's dogma that the amino acid sequence of a protein determines its native structure[43]. The WSME-L model can be modified to predict oxidative folding by considering the stabilization contact energy corresponding to disulfide bond formation when a virtual linker is created between two native Cys pairs (Fig. 1f; see Methods for details). We applied this model, named the WSME-L(SS) model, to three representative proteins for oxidative folding studies: hen egg-white lysozyme (α + β; four disulfide bonds; Fig. 4a), RNase A (α + β; four disulfide bonds; Fig. 4d), and bovine pancreatic trypsin inhibitor (BPTI; small; three disulfide bonds; Fig. 4g). We also calculated the degree of disulfide bond formation between the residues $i$ and $j$ during folding, $\Phi_{i,j}^{SS}$ (see Methods for details). The WSME-L(SS) model predictions described below are consistent with the experimental results and reveal complex behaviors in forming multiple disulfide bonds during folding that were not predicted by the original models (Supplementary Figs. 21–27). Note that although the WSME-L(SS) model cannot simulate non-native disulfide bond formation because it considers only native contacts, experimental studies demonstrate that well-populated intermediates in oxidative folding contain only native disulfide bonds[44], which can be explained by our model.

Lysozyme is one of the best studied multidomain proteins in terms of protein folding. It consists of two domains: a discontinuous α-domain (residues 1–39 and 86–129) and a continuous β-domain (residues 40–85) (Fig. 4a). It has four disulfide bonds: two (6–127 and 30–115) in the α-domain, one (64–80) in the β-domain, and one (76–94) in the interdomain region. Oxidative folding experiments with lysozyme revealed two parallel folding pathways: the major pathway

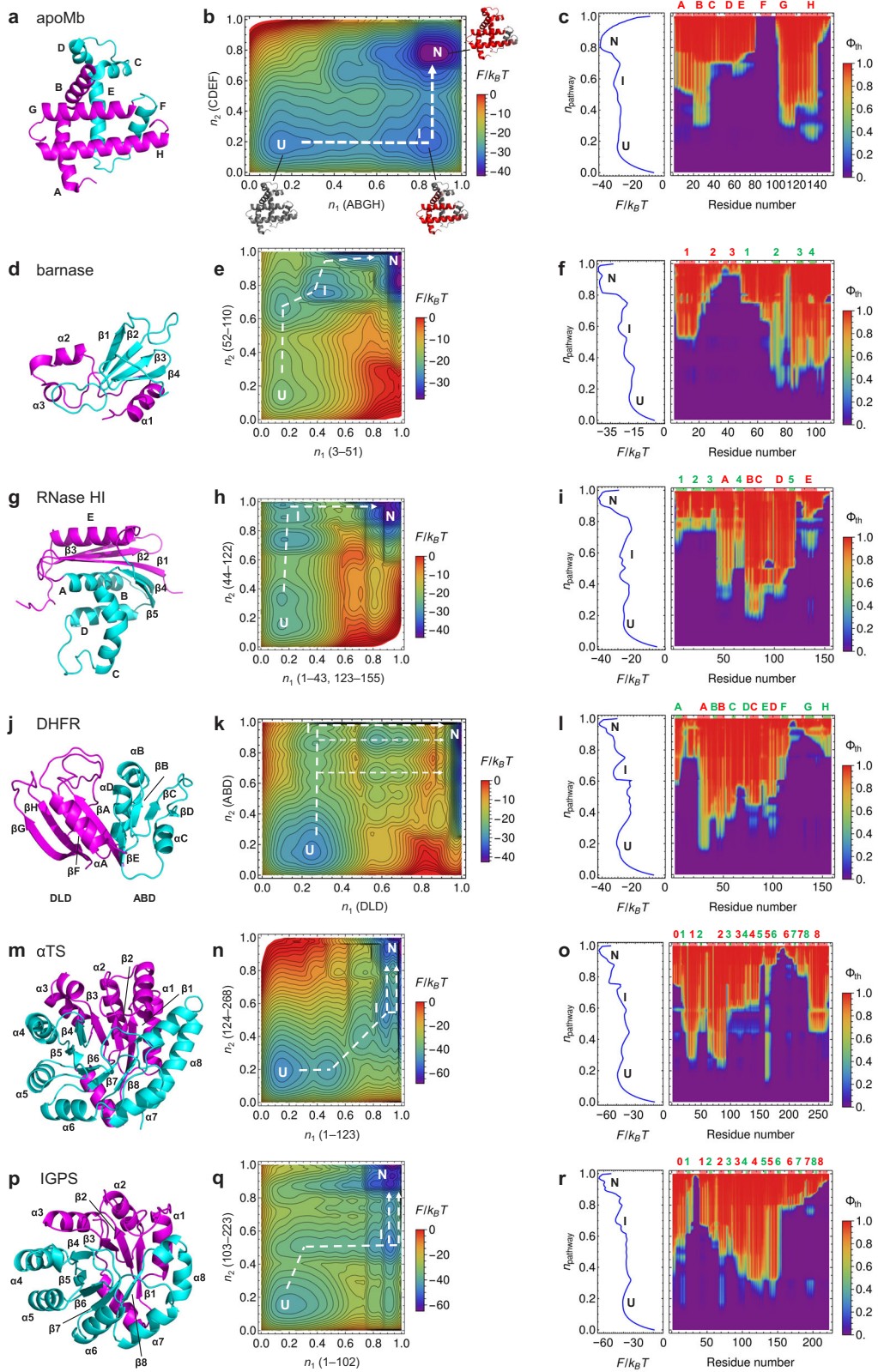

accumulates two intermediates, named des[76–94] and des[64–80], in which three disulfide bonds other than 76–94 and 64–80 are formed, respectively, whereas the minor pathway involves the des[6–127] intermediate (Supplementary Fig. 21a)[45]. The WSME-L(SS) model predicted that the oxidative folding of lysozyme starts with forming the 64–80 disulfide bond in the β-domain and proceeds through two contrasting pathways (Fig. 4b, c). In the major pathway

(Pathway 1), as the preformed 64–80 disulfide bond is reduced, the 30–115 and then the 6–127 disulfide bonds are formed in the α-domain. Subsequently, during the repeated formation and reduction of the 76–94 disulfide bond, the 64–80 and finally the 76–94 disulfide bonds were formed in the β-domain, corresponding to the des[76–94] and des[64–80] intermediates observed in the experiments. In contrast, the minor pathway (Pathway 2) favored the

**Fig. 3 | Folding of large multidomain proteins predicted by WSME-L model.**
**a** Native structure of myoglobin consisting of A, B, G, and H helices (magenta) and
C, D, E, and F helices (cyan). **b** Two-dimensional (2D) free energy landscape of
apomyoglobin (apoMb). $n_1$ and $n_2$ are order parameters for magenta and cyan
regions in (**a**), respectively. Dominant folding pathway is indicated by white dashed
line. Residues predicted to be folded by theoretical Φ-value analysis are shown in
red for structures of intermediate and native state. U, I, and N denote unfolded,
intermediate, and native states, respectively. **c** Residue-specific structure formation
of apoMb by theoretical Φ-value analysis. The $Φ_{th}$-values along dominant folding
pathway (order parameter $n_{pathway}$) are plotted against residue number. Cross-
section of 2D free energy landscape along the pathway is shown on left. Red and
green boxes in upper frame indicate locations of helices and strands, respectively,
and their names are shown in the corresponding colors. **d–f** Results for barnase. In

(**d**), N-terminal (residues 3–51) and C-terminal (residues 52–110) halves are shown in
magenta and cyan, respectively. **g–i** Results for ribonuclease HI (RNase HI). In (**g**),
residues 1–43 and 123–155 are shown in magenta, and residues 44–122 are shown in
cyan. **j–l** Results for dihydrofolate reductase (DHFR). In (**j**), discontinuous loop
domain (DLD; residues 1–37 and 107–159) and adenosine-binding domain (ABD,
residues 38–106) are shown in magenta and cyan, respectively. **m–o** Results for α-
subunit of tryptophan synthase (αTS). In (**m**), N-terminal (residues 1–123) and
C-terminal (residues 124–268) halves are shown in magenta and cyan, respectively.
**p–r** Results for indole-3-glycerol phosphate synthase (IGPS). In (**p**), N-terminal
(residues 1–102) and C-terminal (residues 103–223) halves are shown in magenta
and cyan, respectively. Details are as described in (**a–c**). Source data are provided as
a Source data file.

formation of the 64–80 and then the 76–94 disulfide bonds in the β-
domain, followed by the formation of the 30–115 and then the 6–127
disulfide bonds in the α-domain. This corresponds to the des[6–127]
intermediate observed experimentally.

The prediction of the oxidative folding of RNase A by the WSME-
L(SS) model agrees with the experimental results (Supplementary
Fig. 21b)[46] as follows: in the early stages of folding, the 65–72 disulfide
bond was strongly favored among the four native disulfide bonds
(Fig. 4f). RNase A folded along two parallel pathways (Fig. 4e). In
Pathway 1, the initially formed 65–72 disulfide bond was broken in the
middle stage and regenerated in the late stage of folding (Fig. 4f).
In Pathway 2, a 40–95 disulfide bond was formed in the final
step (Fig. 4f).

Surprisingly, the WSME-L(SS) model successfully reproduced the
oxidative folding experiments of BPTI under conditions where the
glutathione concentrations are close to those in the endoplasmic
reticulum (Fig. 4h, i and Supplementary Fig. 21c)[47,48]. The predictions
showed that BPTI has a high propensity to form a single disulfide
intermediate, named [14–38], with a disulfide bond between residues
14 and 38 early in the folding reaction, but in the middle stage of
folding ($n > 0.4$) it forms the [30–51] intermediate, followed by a two-
disulfide [30–51; 14–38] intermediate (Fig. 4i). BPTI then folds into its
native state via two pathways, both involving the additional formation
of a 5–55 disulfide bond (Fig. 4h, i). Interestingly, Pathway 1 predicted
the partial breakage of the 14–38 disulfide bond before the formation
of all the disulfide bonds. This is consistent with the experimental
results observed at low concentrations of oxidized glutathione, where
the 14–38 disulfide bond is transiently reduced before forming the
native state[44].

## Folding of disulfide-intact proteins

Because oxidative folding measurements have limited temporal reso-
lution, many experimental studies of protein folding have been per-
formed on disulfide-intact proteins, in which all disulfide bonds are
preformed prior to folding experiments. This enables a detailed
investigation of folding kinetics with high temporal resolution using
various experimental methods. To predict the folding reactions of
disulfide-intact proteins, we constructed the WSME-L(SS$_{intact}$) model,
which allows WSME-L predictions in the presence of covalent linker(s)
(Fig. 1g; see Methods for details). We applied this model to predict the
folding of disulfide-intact lysozyme, RNase A, and BPTI.

Kinetic folding experiments on disulfide-intact lysozyme
revealed a complex folding behavior involving a collapsed inter-
mediate and two subsequent parallel pathways (Fig. 5a and Supple-
mentary Fig. 28a)[49–52]. The major slow folding pathway accumulates
the α-domain intermediate in which the discontinuous α-domain is
formed but not the continuous β-domain, whereas in the minor fast
folding pathway the collapsed intermediate folds directly to the
native state with simultaneous formation of α- and β-domains. This
behavior was not explained by the original WSME models, which
predicted that the α-domain was formed only after the formation of

the β-domain (Supplementary Figs. 29 and 30). Strikingly, the WSME-
L(SS$_{intact}$) model predicted the presence of two intermediates (I$_1$ and
I$_2$) and two dominant folding pathways, in agreement with the
experimental results (Fig. 5b–d). In the I$_1$ intermediate, the B- and
D-helices in the α-domain are partially formed, while in the I$_2$ inter-
mediate in Pathway 1, the α-domain is mostly folded but the β-
domain is not (Fig. 5b, c). In contrast, in Pathway 2, after the forma-
tion of I$_1$, the β-domain and then the rest of the α-domain are formed
through a downhill landscape without accumulation of stable inter-
mediates (Fig. 5b, d). These predictions are consistent with the
experimental results and provide detailed insights into the residue-
specific structure formation during lysozyme folding.

Kinetic analysis based on the 2D free energy landscape predicted
that 91% and 9% of lysozyme molecules fold through Pathways 1 and 2,
respectively (Fig. 5e; see Methods for details). Pathway 1 is favored due
to a lower free energy barrier (TS$_1$) than Pathway 2 (TS$_3$) (Fig. 5b).
However, folding via Pathway 1 is slower because it has to overcome a
rate-limiting free energy barrier (TS$_2$) to reach the native state (Fig. 5b,
c, e). This provides a rationale for the experimental observations of a
major but slow folding pathway and a minor but fast folding pathway.
In addition, the time evolution of α- and β-domain structure formation,
calculated by combining the kinetic and theoretical Φ-value analyses,
showed that α-domain folding precedes β-domain folding, reprodu-
cing the experimental results (Fig. 5g).

Thermodynamic analysis predicted that the I$_2$ intermediate is
destabilized at high temperatures (Fig. 5f and Supplementary
Figs. 31–35), consistent with experimental results showing that the
population of the α-domain intermediate is reduced at high
temperatures[53]. Furthermore, the temperature-dependent changes in
the heat capacity of lysozyme obtained from equilibrium unfolding
experiments[54] were well reproduced by our model (Fig. 5h).

To further investigate the role of each disulfide bond in deter-
mining the folding pathways, we calculated the free energy land-
scapes of lysozyme variants containing only a single disulfide bond
using the WSME-L(SS$_{intact}$) model (Supplementary Fig. 36). We found
that the disulfide bonds in the α-domain, especially the 30–115 dis-
ulfide bond, are critical for making Pathway 1 the major folding
pathway (Supplementary Fig. 36). This finding is consistent with
experimental studies showing that the two disulfide bonds in the α-
domain are essential for the accumulation of the α-domain inter-
mediate (corresponding to I$_2$)[55].

We also applied the WSME-L(SS$_{intact}$) model to predict the folding
of disulfide-intact RNase A and BPTI (Supplementary Figs. 37–40).
RNase A has two *cis* Pro residues (Pro93 and Pro114) in its native state
and shows very fast folding reactions when starting from the U$_{vf}$
unfolded state in which both Pro residues are in *cis* conformations
(Supplementary Fig. 28b)[56]. The model predicted the formation of
helix 2 and the N-terminal side of the β-sheet in the intermediate, but
not residues 50–75 and the C-terminal side of the β-sheet (Supple-
mentary Figs. 37 and 38). In addition, stabilization of the entire helix 1
occurs only in the final step. These predictions well reproduce the

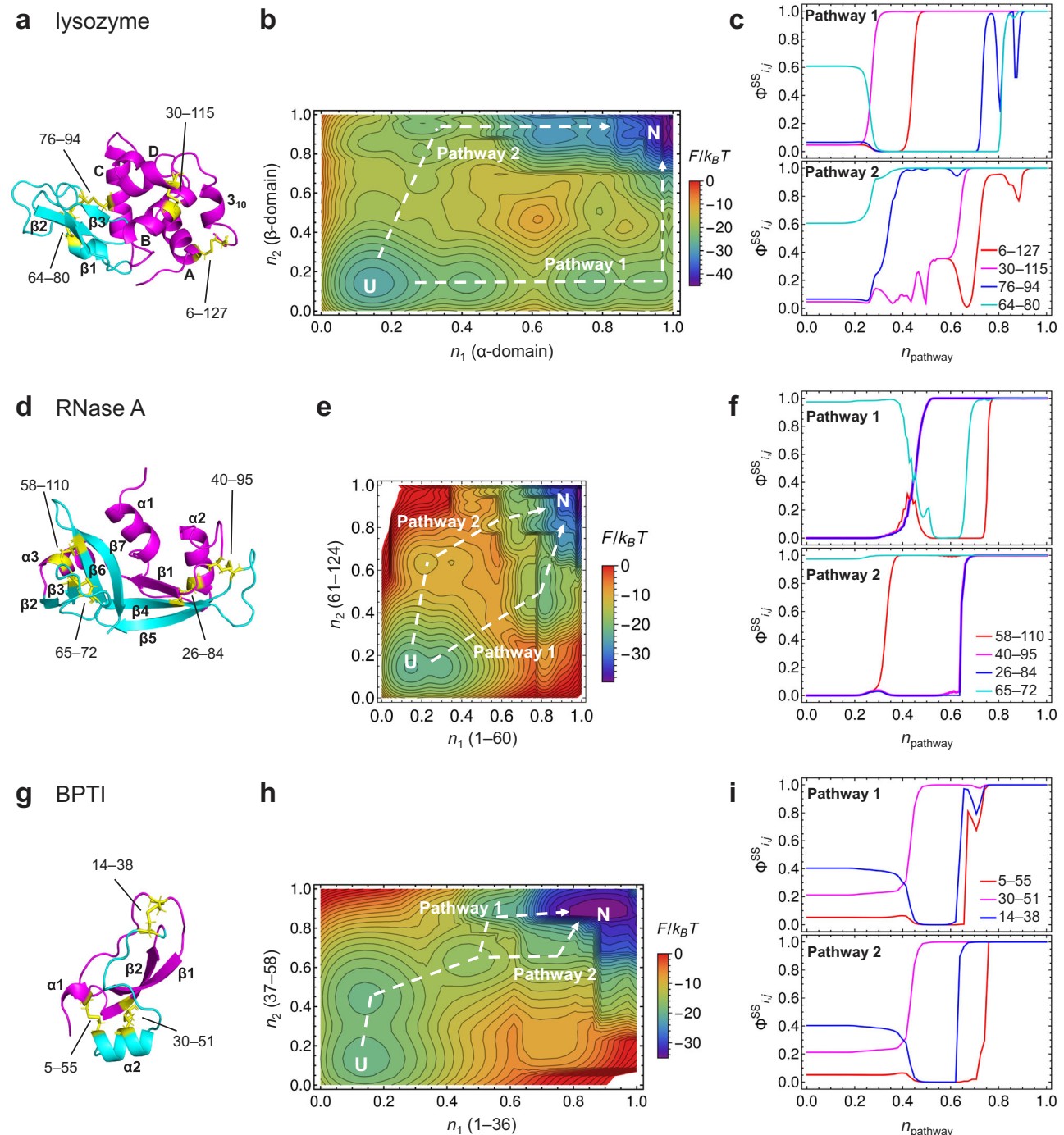

**Fig. 4 | Folding with oxidative disulfide bond formation predicted by WSME-L(SS) model. a** Native structure of lysozyme. α-domain (residues 1–39 and 86–129) and β-domain (residues 40–85) are shown in magenta and cyan, respectively. Disulfide bonds are shown in yellow with Cys residue numbers. **b** Two-dimensional free energy landscape of lysozyme predicted by WSME-L model. $n_1$ and $n_2$ are order parameters for magenta and cyan regions in (**a**), respectively. Dominant folding pathways are indicated by white dashed lines. **c** Degree of disulfide bond formation $\Phi_{i,j}^{SS}$ along Pathways 1 (top) and 2 (bottom). **d–f** Results for RNase A. In (**d**), N-terminal (residues 1–60) and C-terminal (residues 61–124) halves are shown in magenta and cyan, respectively. **g–i** Results for bovine pancreatic trypsin inhibitor (BPTI). In (**g**), N-terminal (residues 1–36) and C-terminal (residues 37–58) regions are shown in magenta and cyan, respectively. Details are as described in (**a–c**). Source data are provided as a Source data file.

structure of the $I_\Phi$ intermediate that accumulates in the folding reaction from $U_{vf}$[56]. For disulfide-intact BPTI, the model predicted a folding pathway similar to oxidative folding; however, the presence of the intact disulfide bonds stabilized the protein, resulting in a downhill folding with a minimal transition state barrier (Supplementary Figs. 39 and 40). Since kinetic folding experiments for disulfide-intact BPTI are not available, this prediction requires experimental verification.

## Discussion

In this study, we developed a simple structure-based theoretical model (WSME-L) to predict protein folding mechanisms by introducing virtual linkers at arbitrary positions to enhance nonlocal interactions. Applying the original WSME models to predict folding processes has been limited to small proteins and multidomain proteins with close N- and C-termini, such as DHFR[12,13,23]. In contrast, our model can be

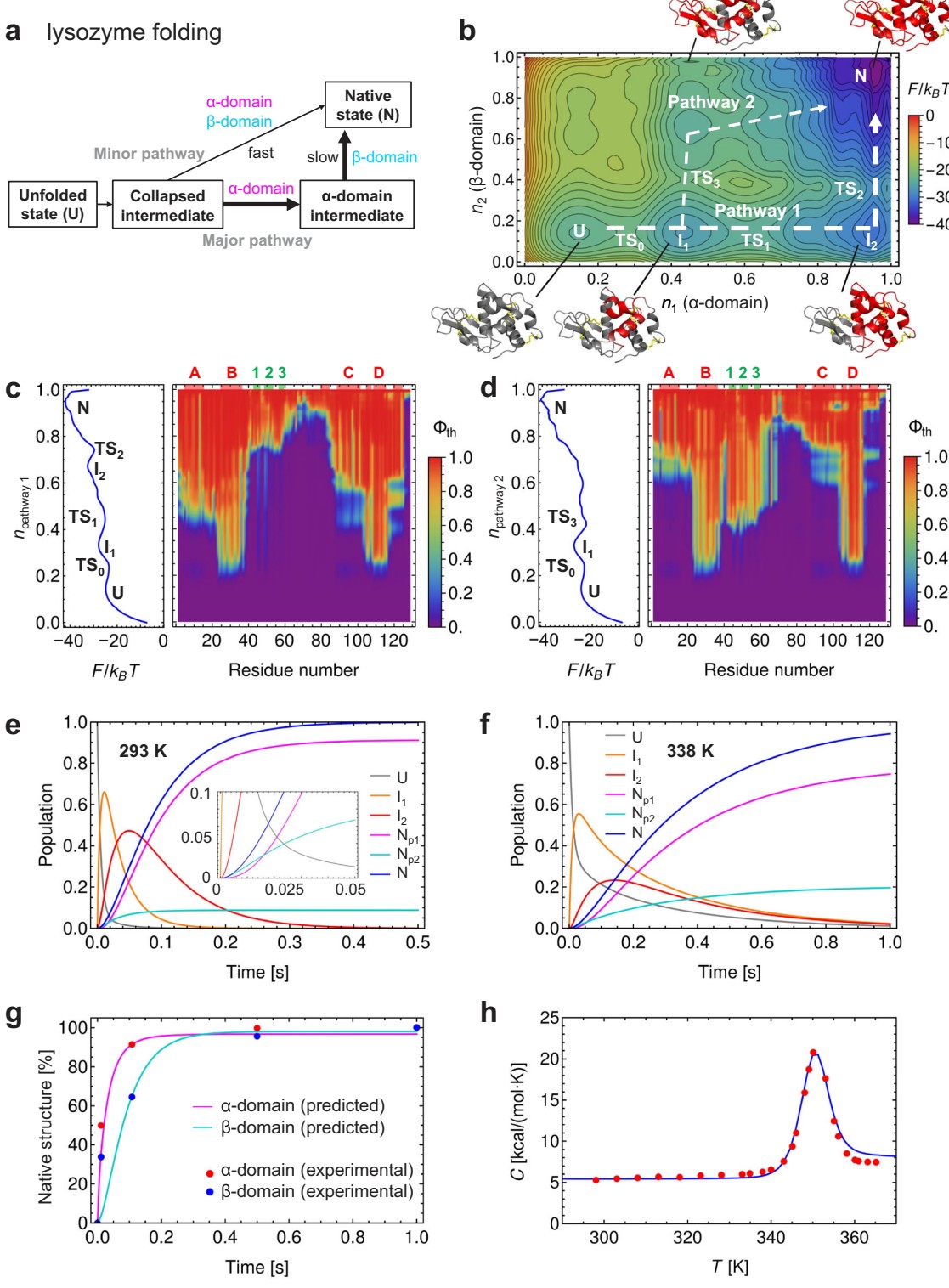

applied to a wide variety of proteins, regardless of size and shape. For small single-domain proteins, the WSME-L model outperformed the original models in terms of prediction accuracy. Furthermore, the WSME-L, WSME-L(SS), and WSME-L(SS$_{intact}$) models successfully predicted the folding pathways of large multidomain proteins, folding with oxidative disulfide bond formation, and folding of disulfide-intact proteins, respectively. Notably, the results are consistent with the experimental results, a feat that was not achieved with the original models. The success of these predictions substantiates the importance

of nonlocal interactions in predicting folding mechanisms, particularly for multidomain proteins[17,23,27,57]. In addition, the models yielded detailed predictions of folding processes beyond reproducing the experimental results and provided the key determinants and rationale for the folding mechanisms.

An important assumption of the WSME model is that a protein is stabilized only by the contacts present in the native structure. The same assumption is employed in an ideal protein based on the consistency principle proposed by Gō[58] and in the perfect funnel-like

**Fig. 5 | Folding of disulfide-intact lysozyme predicted by WSME-L(SS_intact) model. a** Folding mechanism of experimentally elucidated hen egg-white lysozyme[49–52]. **b** Two-dimensional (2D) free energy landscape of lysozyme predicted by WSME-L model under folding conditions at 293 K. $n_1$ and $n_2$ are order parameters for α- and β-domains, respectively. Folding pathways 1 and 2 are indicated by white dashed lines. Lysozyme residues predicted to be folded by theoretical Φ-value analysis ($\Phi_{th} > 0.5$) are shown in red for structures of intermediates and native state. U and N denote unfolded and native states, respectively; $I_1$ and $I_2$ denote intermediates; and $TS_0$, $TS_1$, $TS_2$, and $TS_3$ denote transition states. **c, d** Residue-specific structure formation along folding pathways of lysozyme predicted by theoretical Φ-value analysis. $\Phi_{th}$-values along Pathways 1 (**c**) and 2 (**d**) are plotted against residue number. Cross-section of 2D free energy landscape along folding pathway is shown on left. Red and green boxes in upper frame indicate locations of helices and strands, respectively, and their names are shown in the corresponding colors. **e, f** Time evolution of concentrations of kinetic species [U (gray), $I_1$ (orange), $I_2$ (red), and N (blue)] at 293 K (**e**) and 338 K (**f**). $N_{p1}$ (magenta) and $N_{p2}$ (cyan) indicate native state formed through Pathways 1 and 2, respectively. In (**e**) is expanded view of time evolution during first 0.05 s. Total concentration was normalized to 1. **g** Time evolution of each domain. Predictions by WSME-L model (α-domain: magenta curve, and β-domain: cyan curve), and average proton occupancy of each domain obtained from pulsed-hydrogen exchange NMR experiments[49] (α-domain: red filled circles, and β-domain: blue filled circles). **h** Temperature dependence of heat capacity. Blue curve: prediction by WSME-L(SS_intact) model. Red filled circles: experimental data[54]. Source data are provided as a Source data file.

energy landscape based on the principle of minimal frustration proposed by Wolynes, Onuchic, and colleagues[59]. Despite their simplicity, WSME models successfully predict the folding mechanisms of single-domain and multidomain proteins. These results suggest that both the consistency and minimal frustration principles hold for many proteins, regardless of their size. This suggests that real proteins behave similarly to ideal proteins regarding folding kinetics and thermodynamics.

Detailed comparisons of predicted and experimental data reveal slight differences between ideal and real proteins due to non-native interactions and Pro isomerization during folding. In disulfide-intact lysozyme folding, non-native interactions stabilize the α-domain intermediate ($I_2$) and retard the subsequent folding process[51]. The WSME-L prediction underestimated the maximum population of $I_2$ (~45%) compared to the experimental results (~60%)[53] (Fig. 5e), but this discrepancy can be resolved by accounting for the stabilization effect of non-native interactions by 0.44 kcal/mol (Supplementary Fig. 41). In αTS and IGPS folding, off-pathway intermediates formed early in folding by non-native interactions must be disrupted before folding to the native state[41,42]. In addition, *cis* Pro residues in the native state of αTS, IGPS, and RNase A must isomerize from *trans* to *cis* conformations during folding if they are in *trans* conformations in the unfolded state, resulting in the appearance of additional parallel folding pathways[41,42,56]. However, the WSME-L models predicted only the folding pathways involving on-pathway intermediates with native-like Pro conformations because they only consider native contacts. Similarly, the WSME-L(SS) model predicted that the 40–95 disulfide bond is formed together with the 26–84 disulfide bond in the final step of oxidative RNase A folding, whereas experiments have shown that the final step of the folding involves only the formation of the 40–95 disulfide bond[46]. This may be because the slow *trans*-to-*cis* isomerization of Pro93 near the 40–95 disulfide bond[60], which is not considered in the WSME-L(SS) model, delays the formation of this disulfide. Thus, the WSME-L models are useful for capturing the general features of the folding mechanisms of ideal proteins, and slight discrepancies between the predicted and experimental data can provide clues to understanding the role of non-native interactions/conformations in protein folding. Future efforts to incorporate non-native interactions and Pro isomerization into the models would allow for even greater accuracy in predicting protein folding mechanisms.

Theoretical approaches to predicting three-dimensional protein structures have been greatly advanced by deep learning[1,2], making considerable progress toward solving the structure prediction component of the 'protein folding problem'[3]. Although these algorithms do not predict how proteins fold and function, WSME-L models can predict the folding mechanisms of various proteins, without the limitations of size and shape, with low computational complexity, using only native structures solved experimentally or with deep learning algorithms. Moreover, because the WSME-L model can describe protein dynamics in terms of free energy landscapes, it has a wide range of

potential applications beyond protein folding, including the dynamic motions associated with protein function[19,20,22], protein–protein interactions, and coupled folding and binding of intrinsically disordered proteins[61]. Furthermore, it would be applicable to the development of novel protein design methods based on dynamics prediction. Therefore, the WSME-L models may pave the way for solving the folding process component of the 'protein folding problem'[3] and will be increasingly useful in the forthcoming era of computational biology.

## Methods
### Contact energies
The native contact energy between residues $i$ and $j$, $\varepsilon_{i,j}$, was obtained as $\varepsilon_{i,j} = \varepsilon e_{i,j}$, where $\varepsilon$ is the energy size of the inter-residue contacts in a protein and $e_{i,j} (-1 \leq e_{i,j} \leq 1)$ is the weight of the contact energy between residues $i$ and $j$. The $e_{i,j}$ values were determined as follows: first, the three-dimensional structure from the Protein Data Bank (PDB) was energy minimized using AMBER18/ff14SB with restraints for backbone atoms[62]. In Original model 1, $e_{i,j} = -1$ if an atom in residue $i$ is within 4 Å of an atom in residue $j$; otherwise $e_{i,j} = 0$. In Original model 2, using the energy-minimized structure, the AMBER-derived contact energy $\varepsilon_{i,j}^{AMBER}$ was calculated using the MMPBSA.py module for implicit solvents[62]. $e_{i,j}$ was defined as $\varepsilon_{i,j}^{AMBER}$ divided by the maximum absolute value of $\varepsilon_{i,j}^{AMBER}$ among all inter-residue contacts excluding those with neighboring residues ($j \leq i + 2$). The WSME-L model has two types of partition functions: $Z(n)$ and $Z^{(u,v)}(n)$ (see Eq. (7)). The $e_{i,j}$ values in $Z(n)$ were the same as those used in Original model 2. $Z^{(u,v)}(n)$ is described as:

$$Z^{(u,v)}(n) = \mathrm{Tr}_n \exp\left[ -\frac{1}{k_B T} \left( H^{(u,v)}(\{m\}) + \varepsilon'^{(u,v)} - T \sum_{i=1}^{N} S_i m_i \right) \right] \quad (9)$$

In $Z^{(u,v)}(n)$, the contact energy acquired by the virtual linker formation between residues $u$ and $v$, $\varepsilon'^{(u,v)}$, was defined as $\varepsilon'^{(u,v)} = \varepsilon(e_{u,v} + e_{u+1,v} + e_{u-1,v} + e_{u,v+1} + e_{u,v-1})$. The $e_{i,j}$ values for $H^{(u,v)}(\{m\})$ in $Z^{(u,v)}(n)$ were the same as in $Z(n)$, except that $e_{u,v} = e_{u\pm1,v} = e_{u,v\pm1} = 0$ to avoid double counting. For all models, $e_{i,j} = 0$ for neighboring residues ($j \leq i + 2$).

In the predictions of folding with oxidative disulfide bond formation, −40 kcal/mol was added to $\varepsilon_{i,j}^{AMBER}$ for the residue pair forming a disulfide bond in Original model 2 and the WSME-L(SS) model. In Original model 1, the change in contact energy corresponding to the addition of −40 kcal/mol in Original model 2 was added to the contact energy for the disulfide pair. To predict the folding of disulfide-intact proteins using the WSME-L(SS_intact) model, $e_{i,j} = e_{i\pm1,j} = e_{i,j\pm1} = 0$ for the residue pairs $i$ and $j$ forming a disulfide bond.

It is important to note that the use of native contact energies calculated by the AMBER force field was necessary to obtain the free energy landscape consistent with the experimental data. For disulfide-intact lysozyme, the landscape calculated using a uniform contact energy predicted only a single folding pathway corresponding to Pathway 1 (Supplementary Fig. 42). This inconsistency with the

experiments is a marked contrast with previous studies in which residue-specific weighing on contact energies in an Ising-like coarse-grained model had a small contribution to the folding prediction of small proteins[13,57]. These results suggest the importance of side-chain packing interactions in determining the folding mechanisms of multidomain proteins.

The three-dimensional structure used to calculate the contact energies of apoMb was determined as follows: the all-atom molecular dynamics simulations of apoMb with heme removed from the heme-bound structure were performed using GROMACS 2021.2 with ff14SB for 1 μs with explicit solvents, as only heme-bound myoglobin structures were available from PDB[63,64]. The initial structure was built using the LEaP module of the AmberTools19 package with ff14SB[62] and the crystal structure of myoglobin (PDB ID: 1bzp) as input. All ionizable side chains were set to their pH 7 protonation state. Charge-neutralizing chloride ions were placed around the protein molecule. The protein molecule was immersed in a 70.4 × 65.9 × 70.6 Å³ periodic box of TIP3P water molecules (7680 water molecules and 25,501 atoms in total). The long-range electrostatic interactions were treated using the particle-mesh Ewald method. The systems were energy minimized by the steepest descent algorithm for 200 steps with positional restraints and for additional 200 steps without restraints[64]. The system was then heated from 0 K to 310 K during a 200-ps constant-NVT MD simulation with harmonic position restraints on the heavy atoms of the solutes (with a force constant of 10 kcal mol⁻¹ A⁻²). During the subsequent 700-ps constant-NPT MD simulation at 310 K and $1.0 \times 10^5$ Pa, the force constants of the position restraints were gradually reduced. The system was further equilibrated for 100 ps without position restraints. The bonds between hydrogen atoms and heavy atoms were constrained using the P-LINCS algorithm, allowing the use of 2-fs time steps. Temperature was controlled using the stochastic velocity-rescaling (V-rescale) algorithm. Pressure in NPT simulations was controlled using a Berendsen barostat with a coupling constant of 1 ps⁻¹ [64]. The time-course analysis showed that the energy of the system was well converged during equilibration. Subsequently, an unrestrained constant-NPT MD simulation was performed for 1 μs at 310 K and $1.0 \times 10^5$ Pa using a Parrinello-Rahman barostat with a coupling constant of 2 ps⁻¹ [64], and the native ensembles of apoMb were obtained. The MD trajectories were analyzed using the CPPTRAJ module in AmberTools19[62]. The 1-μs MD simulations were performed in triplicate under the same conditions. The three independent MD trajectories represented by a root mean square deviation (RMSD) of the main-chain Cα atoms from the initial structure were similar (Supplementary Fig. 43a), and the average RMSD in each trajectory was almost the same for all trajectories, indicating the reproducibility of the MD simulations (Supplementary Fig. 43b). We then performed cluster analysis on 3000 structures taken every 1 ns from a total of 3-μs simulations with an RMSD cutoff of 1.6 Å[65]. The central structure of the first cluster was used as a computational model of apoMb to calculate contact energies; the PDB file of the apoMb structure is provided as Supplementary Data 1. As the F helix of apoMb does not form a stable structure in the native state[66], the contact energies involving the residues in the F helix were set to zero. The resulting free energy landscape was consistent with the experimentally observed folding mechanisms of apoMb (Fig. 3b, c), indicating that both the accuracy and timescale of the MD simulation are sufficient for the present study.

### Entropic costs

The entropic cost of the main chain of the $i$-th residue, $S_i$, was set to −2.0 cal/(mol·K) for Original model 1, −2.5 cal/(mol·K) for Original model 2, and −3.5 cal/(mol·K) for the WSME-L models.

The entropic cost of ring closure via virtual linker formation, $S'^{(u,v)}(n)$, was calculated as follows: let us consider a Gaussian chain

with a chain length of $La$, where $L$ is a natural number, and $a$ (3.8 Å) is the distance between two adjacent Cα atoms. The entropic cost of ring closure between the two termini of this chain is:

$$s'(L) = -\frac{3}{2} k_B \left( \ln L + \frac{r^2 - a^2}{2AaL} \right) \quad (10)$$

where $A$ (20 Å) is the persistence length of a peptide chain, and $r$ is the distance between the Cα atoms of the two termini of the chain[23]. Using this, we defined $S'^{(u,v)}(n)$ as the arithmetic mean of the entropic cost $s'$ for all possible states as follows:

$$S'^{(u,v)}(n) = h_{S'} \sum_{i=0}^{nN} \frac{\binom{N'}{i}\binom{N-N'}{nN-i}}{\binom{N}{nN}} s'(N' - i) \quad (11)$$

where $N' = v - u + 1$, $h_{S'}$ is a scaling factor of the ring entropy, and () denotes a combination.

### Folding rate and stability of small proteins

The folding rate and stability of small proteins were calculated from the 1D free energy landscape according to previous studies[67]. The microscopic rate constant for the transition from a structure with $nN$ native contacts (order parameter $n$) to one with $nN \pm 1$ native contacts (order parameter $n \pm 1/N$) can be described by:

$$k_{nN,nN\pm1} = A \exp\left( -\frac{F(n \pm 1/N) - F(n)}{2k_B T} \right) \quad (12)$$

where $A = 10^7$ and $F(n)$ is the 1D free energy at order parameter $n$. The macroscopic rate constant was obtained as the eigenvalue with the smallest nonzero absolute value of the following rate matrix:

$$\begin{pmatrix} -k_{0,1} & k_{1,0} & 0 & \cdots & 0 & 0 \\ k_{0,1} & -k_{1,0}-k_{1,2} & k_{2,1} & \cdots & 0 & 0 \\ 0 & k_{1,2} & -k_{2,1}-k_{2,3} & \cdots & 0 & 0 \\ \vdots & \vdots & \vdots & \ddots & \vdots & \vdots \\ 0 & 0 & 0 & \cdots & -k_{N-1,N-2}-k_{N-1,N} & k_{N,N-1} \\ 0 & 0 & 0 & \cdots & k_{N-1,N} & -k_{N,N-1} \end{pmatrix}$$

$$(13)$$

Under conditions in which the native state is sufficiently stable relative to the unfolded state, the macroscopic rate constant is equivalent to the folding rate constant $k_f$.

The stability of the small proteins, $\Delta G_{NU}$, was estimated as the difference between the minimum free energy values of the native and unfolded state basins in a 1D free energy landscape at 293 K or 298 K.

### Determination of parameters

In the present WSME-L models, the parameters to be determined for each protein are $\varepsilon$ and $h_{S'}$. To determine these values for small proteins, the $h_{S'}$ was set to 0.5–2.0 in increments of 0.1, and for each of them, the $\varepsilon$ was determined that minimized the following loss function for the stability and folding rate between the experimentally determined and predicted values:

$$loss = \left( \frac{\Delta G_{NU}}{k_B T} - \frac{\Delta G_{NU}^{exp}}{k_B T} \right)^2 + \left[ \ln\left( \frac{k_f}{k_f^{exp}} \right) \right]^2 \quad (14)$$

Among the pairs of $h_{S'}$ and $\varepsilon$ thus obtained, we selected a pair for which the theoretical Φ-values correlate best with the experimental values. Similarly, for other proteins, we searched for the pairs of $h_{S'}$ and $\varepsilon$ that

minimized the loss function for the stability between the experimentally determined and predicted values:

$$loss = \left( \frac{\Delta G_{NU}}{k_B T} - \frac{\Delta G_{NU}^{exp}}{k_B T} \right)^2 \quad (15)$$

Among them, we selected a pair for which the theoretical $\Phi$-values and stability of folding intermediates correlate best with the experimental values. See Supplementary Table 2 for the $\varepsilon$ and $h_{S'}$ values used for each protein.

## Exact solution for WSME-L model

The exact solution of $Z^{(u,v)}(n)$ (see Eq. (9)) was obtained by rearranging the exact solution of the original WSME model[18]. First, we considered the Boltzmann weight of a native stretch, as follows:

$$w_{i,j} = \exp\left[ -\frac{1}{k_B T} \left( \sum_{k=i}^{j-1} \sum_{l=k+1}^{j} \varepsilon_{k,l} - T \sum_{k=i}^{j} S_k \right) \right] \lambda^{j-i+1} \quad (16)$$

where $\lambda$ is an indeterminant whose exponent is related to the order parameter $n$. For convenience in later calculations, we define $w_{i+1,i} = 1$. To reduce computational complexity, virtual linkers were introduced only for residues $i$ and $j$ with interaction energies $\varepsilon_{i,j}^{AMBER}$ less than –0.6.

According to the exact solution of the original WSME model, the generating function of the partition function $Z$, restricted by an order parameter $n$, is written as:

$$G_Z(Z(n); \lambda) = {}^0\langle 0 | \prod_{i=0}^{N} Q_{i+1}^i | 0 \rangle^{N+1} \quad (17)$$

Here, $Q_{\mu+1}^\mu$ was a transfer matrix defined as:

$$Q_{\mu+1}^\mu = \sum_{k=1}^{\mu+1} |k-1\rangle^{\mu\,\mu+1}\langle k| + \sum_{k=0}^{\mu} w_{\mu-k+1,\mu} |k\rangle^{\mu\,\mu+1}\langle 0| \quad (18)$$

where $|k\rangle^\mu$ is a $(\mu+1)$-dimensional vector whose set satisfies the orthonormal system:

$$\begin{cases} {}^\mu\langle k|k'\rangle^\mu = \delta_{k,k'} \\ \sum_{k=0}^{\mu} |k\rangle^{\mu\,\mu}\langle k| = I_{\mu+1} \end{cases} \quad (19)$$

where $k$, $k' = 0, 1, \cdots, \mu$. From the generating function, the partition function restricted by the order parameter is formally given as follows:

$$Z(n) = \frac{1}{(nN)!} \left( \frac{\partial}{\partial \lambda} \right)^{(nN)} G_Z \Big|_{\lambda=0} \quad (20)$$

When calculating high-dimensional free energy landscapes, more indeterminants with proper exponents should be prepared similarly.

Next, when a linker is provided, the additional weights corresponding to the interactions between the two native stretches connected by the linker should be considered. The introduction of a linker connecting residues $u$ and $v$ implied that an additional weight, $w_L^{\alpha,\beta,\gamma,\delta}$, should be multiplied when calculating the products of $w_{\alpha,\beta}$ ($\alpha \le u \le \beta$) and $w_{\gamma,\delta}$ ($\gamma \le v \le \delta$). The additional weight is given as:

$$w_L^{\alpha,\beta,\gamma,\delta} = \exp\left( -\frac{1}{k_B T} \sum_{k=\alpha}^{\beta} \sum_{l=\gamma}^{\delta} \varepsilon_{k,l} \right) \quad (21)$$

For convenience in later calculations, we defined that $w_L^{\alpha,\beta,\gamma,\delta} = 1$ when $u < \alpha$ or $\beta < u$.

The products of $w_{\alpha,\beta}$ and $w_{\gamma,\delta}$ must be extracted to calculate the partition function using additional weights. To achieve this, we insert a unit matrix into the generating function as follows:

$$G_Z(Z(n); \lambda) = {}^0\langle 0 | \left( \prod_{i=0}^{v-1} Q_{i+1}^i \right) \left( \sum_{k=0}^{v} |k\rangle^{v\,v}\langle k| \right) \left( \prod_{i=v}^{N} Q_{i+1}^i \right) | 0 \rangle^{N+1} \quad (22)$$

By rearranging this expression according to the definition of the transfer matrix, the generating function can be expanded as:

$$G_Z(Z(n); \lambda) = \sum_{\gamma=1}^{v} \sum_{\delta=v}^{N} R^{0,\gamma-1} w_{\gamma,\delta} R^{\delta+1,N+1} + R^{0,v} R^{v,N+1} \quad (23)$$

with:

$$R^{i,j} = {}^i\langle 0 | \prod_{k=i}^{j-1} Q_{k+1}^k | 0 \rangle^j \quad (24)$$

where $\gamma = v - k + 1$. Index $\gamma$ runs from 1 to $v$, and index $\delta$ runs from $v$ to $N$. This means we can extract all the weights, including for the $v$-th residue (i.e., $w_{\gamma,v}$ and $w_{v,\delta}$).

We then obtained $R_{\gamma,\delta}^{0,\gamma-1}$ by converting all the weights, $w_{i,j}$, included in $R^{0,\gamma-1}$ into $w_{i,j} w_L^{\alpha,\beta,\gamma,\delta}$. This allowed us to calculate the generating function with a linker, $G_{Z^{(u,v)}}$, which is given by:

$$G_{Z^{(u,v)}}(Z^{(u,v)}(n); \lambda) = \left( \sum_{\gamma=1}^{v} \sum_{\delta=v}^{N} R_{\gamma,\delta}^{0,\gamma-1} w_{\gamma,\delta} R^{\delta+1,N+1} + R^{0,v} R^{v,N+1} \right) \exp\left( -\frac{\varepsilon'^{(u,v)}}{k_B T} \right) \quad (25)$$

The contact maps and free energy landscapes were drawn using Mathematica 12.2 (Wolfram, Champaign, IL, USA).

## Theoretical $\Phi$-value analysis for WSME-L model

In protein folding experiments, residue-specific structure formation has been extensively studied using $\Phi$-value analysis, which requires the measurement of free energy changes in intermediates and transition states due to an amino acid substitution at the residue of interest[68]. Here, we performed a theoretical $\Phi$-value analysis by introducing a small perturbation in the contact energies of the $l$-th residue, equivalent to an amino acid substitution, and calculated the free energy landscape, which corresponded to the kinetic folding measurements of the mutant. The difference in free energy between the perturbed and unperturbed landscapes along a folding pathway was normalized by the total free energy change due to the perturbation, resulting in theoretical $\Phi$-values ($\Phi_{th,l}$): $\Phi_{th,l} = 0$ or 1 when the $l$-th residue was unfolded or fully folded, respectively. The $\Phi_{th,l}$ values in the intermediate or transition states corresponded to the experimentally observed $\Phi$-values.

A theoretical $\Phi$-value analysis of the original WSME model has been performed elsewhere[13]. To apply this to the WSME-L model, we defined the perturbations at the $l$-th amino acid residue as follows:

$$\begin{cases} \Delta H_l = \sum_{i=1}^{l} \eta_{i,l} m_{i,l} + \sum_{j=l}^{N} \eta_{l,j} m_{l,j} \\ \Delta H_l^{(u,v)} = \sum_{i=1}^{l} \eta_{i,l} \lceil (m_{i,l} + m_{i,l}^{(u,v)})/2 \rceil + \sum_{j=l}^{N} \eta_{l,j} \lceil (m_{l,j} + m_{l,j}^{(u,v)})/2 \rceil \end{cases} \quad (26)$$

where $\eta_{i,l}$ and $\eta_{l,j}$ are the contact energy changes due to small perturbations in the native contacts involving the $l$-th residue. All interactions formed by the $l$-th residue were simultaneously modulated; $\eta_{i,j} = 0.1|\varepsilon_{i,j}|$ for $\varepsilon_{i,j} < 0$, while $\eta_{i,j} = 0$ for $\varepsilon_{i,j} > 0$.

The partition function with perturbations owing to such a pseudo-mutation was described as follows:

$$Z_{L,l}^{Mu}(n) = Z_l^{Mu}(n) + \sum_{(u,v):\text{All contacts}} \left( Z_l^{Mu(u,v)}(n) - Z_l^{Mu}(n) \right) \exp\left( \frac{S'^{(u,v)}(n)}{k_B} \right)$$

(27)

where Mu denotes mutation at the $l$-th residue,

$$Z_l^{Mu}(n) = \text{Tr}_n \exp\left[ -\frac{1}{k_B T} \left( H + \Delta H_l - T \sum_{i=1}^N S_i m_i \right) \right]$$

(28)

$$Z_l^{Mu(u,v)}(n) = \text{Tr}_n \exp\left[ -\frac{1}{k_B T} \left( H^{(u,v)} + \Delta H_l^{(u,v)} + \varepsilon'^{(u,v)} + \Delta \varepsilon'^{(u,v)}_l - T \sum_{i=1}^N S_i m_i \right) \right]$$

(29)

and:

$$\Delta \varepsilon'^{(u,v)}_l = \begin{cases} \eta_{u\pm1,v} & (l = u \pm 1) \\ \eta_{u,v-1} + \eta_{u,v} + \eta_{u,v+1} & (l = u) \\ \eta_{u,v\pm1} & (l = v \pm 1) \\ \eta_{u-1,v} + \eta_{u,v} + \eta_{u+1,v} & (l = v) \\ 0 & (\text{otherwise}) \end{cases}$$

(30)

$\Phi_{th,l}$ is obtained as follows:

$$\Phi_{th,l}(n) = \frac{-k_B T \ln[Z_{L,l}^{Mu}(n)/Z_{L,l}^{Mu}(0)] - (-k_B T \ln[Z_L(n)/Z_L(0)])}{-k_B T \ln[Z_{L,l}^{Mu}(1)/Z_{L,l}^{Mu}(0)] - (-k_B T \ln[Z_L(1)/Z_L(0)])}$$

(31)

When the value is less than zero, $\Phi_{th,l}$ is set to zero.

$Z_l^{Mu(u,v)}(n)$ was obtained using the exact solution of the WSME-L model. First, to consider the change in the free energy of a native stretch including the $l$-th residue connected along the main chain, all weights $w_{i,j}$ in the transfer matrices were replaced by $w_{i,j} \exp[-(\sum_{i\le s \le l}\eta_{s,l} + \sum_{l\le t \le j}\eta_{l,t})/k_B T]$. Second, to consider the change in free energy of a native stretch connected via a linker to another native stretch including the $l$-th residue, all weights $w_L^{\alpha,\beta,\gamma,\delta}$ in the transfer matrices were replaced by $w_L^{\alpha,\beta,\gamma,\delta} \exp[-(\sum_{\gamma \le t \le \delta}\eta_{l,t})/k_B T]$ when $\alpha \le l \le \beta$ or by $w_L^{\alpha,\beta,\gamma,\delta} \exp[-(\sum_{\alpha \le s \le \beta}\eta_{s,l})/k_B T]$ when $\gamma \le l \le \delta$. This modification of the weights allowed us to calculate $Z_l^{Mu(u,v)}(n)$ efficiently.

Among the experimentally determined $\Phi$-values, we used the data of the mutants with a destabilization free energy $\Delta\Delta G_{NU}^{exp}$ of more than 0.7 kcal/mol by amino acid substitution to compare the predictions with those of the experiments[68].

**Degree of disulfide bond formation in WSME-L(SS) model**
To calculate the degree of disulfide bond formation between residues $i$ and $j$ during folding, $\Phi_{i,j}^{SS}(n)$, we defined the perturbation at the contact between residues $i$ and $j$ that form a disulfide bond as follows:

$$\begin{cases} \Delta H_{i,j} = \eta_{i,j} m_{i,j} \\ \Delta H_{i,j}^{(u,v)} = \eta_{i,j} \lceil (m_{i,j} + m_{i,j}^{(u,v)})/2 \rceil \end{cases}$$

(32)

Similar to the calculation of $\Phi_{th}$-values, the partition function with the perturbation is described by:

$$Z_{L,i,j}^{Mu}(n) = Z_{i,j}^{Mu}(n) + \sum_{(u,v):\text{All contacts}} \left( Z_{i,j}^{Mu(u,v)}(n) - Z_{i,j}^{Mu}(n) \right) \exp\left( \frac{S'^{(u,v)}(n)}{k_B} \right)$$

(33)

where:

$$Z_{i,j}^{Mu}(n) = \text{Tr}_n \exp\left[ -\frac{1}{k_B T} \left( H + \Delta H_{i,j} - T \sum_{i=1}^N S_i m_i \right) \right]$$

(34)

$$Z_{i,j}^{Mu(u,v)}(n) = \text{Tr}_n \exp\left[ -\frac{1}{k_B T} \left( H^{(u,v)} + \Delta H_{i,j}^{(u,v)} + \varepsilon'^{(u,v)} + \Delta \varepsilon'^{(u,v)}_{i,j} - T \sum_{i=1}^N S_i m_i \right) \right]$$

(35)

and:

$$\Delta \varepsilon'^{(u,v)}_{i,j} = \begin{cases} \eta_{u,v} & (i = u, j = v) \\ \eta_{u\pm1,v} & (i = u \pm 1, j = v) \\ \eta_{u,v\pm1} & (i = u, j = v \pm 1) \\ 0 & (\text{otherwise}) \end{cases}$$

(36)

$Z_{i,j}^{Mu(u,v)}(n)$ was obtained using the exact solution of the WSME-L model. $\Phi_{i,j}^{SS}(n)$ was calculated as follows:

$$\Phi_{i,j}^{SS}(n) = \frac{-k_B T \ln Z_{L,i,j}^{Mu}(n) - (-k_B T \ln Z_L(n))}{\eta_{i,j}}$$

(37)

**Modification of WSME-L model for WSME-L(SS$_{intact}$) model**
The partition function for disulfide-intact folding, $Z_L^{SSintact}(n)$, was constructed by slightly modifying $Z_L(n)$ for the WSME-L model. First, the Hamiltonian with two linkers at the two residue pairs $(u_1, v_1)$ and $(u_2, v_2)$ was given by:

$$H^{(u_1,v_1),(u_2,v_2)}(\{m\}) = \sum_{i=1}^{N-1} \sum_{j=i+1}^N \varepsilon_{i,j} \left\lceil \frac{m_{i,j} + m_{i,j}^{(u_1,v_1)} + m_{i,j}^{(u_2,v_2)}}{3} \right\rceil$$

(38)

The partition function with double linkers,

$$Z^{(u_1,v_1),(u_2,v_2)}(n) = \text{Tr}_n \exp\left[ -\frac{1}{k_B T} (H^{(u_1,v_1),(u_2,v_2)}(\{m\}) + \varepsilon'^{(u_1,v_1)} - T \sum_{i=1}^N S_i m_i) \right]$$

(39)

can be computed using the same solution as for the partition function with a single linker, $Z^{(u,v)}(n)$, by expanding the generating function with the insertion of two unit matrices, $\sum_{k=0}^{v_1} |k\rangle^{v_1}\,^{v_1}\langle k|$ and $\sum_{k=0}^{v_2} |k\rangle^{v_2}\,^{v_2}\langle k|$, as in Eq. (22), and by introducing an additional weight to the weight stabilized by the linker. Using this equation, the partition function for the disulfide-intact folding was defined as follows:

$$Z_L^{SSintact}(n) = \sum_{(u_2,v_2):\text{SS bonds}} \left[ Z^{(u_2,v_2)}(n) + \sum_{(u_1,v_1):\text{All contacts}} \left( Z^{(u_1,v_1),(u_2,v_2)}(n) - Z^{(u_2,v_2)}(n) \right) \exp\left( \frac{S'^{(u_1,v_1)}(n)}{k_B} \right) \right]$$

(40)

where the summation of $(u_2, v_2)$ was calculated for all residue pairs that formed a disulfide bond in a protein. For example, $(u_2, v_2) = (6, 127)$, $(30, 115)$, $(76, 94)$, and $(64, 80)$ for lysozyme, which has disulfide bonds at 6–127, 30–115, 76–94, and 64–80. In contrast, the summation of $(u_1, v_1)$ was calculated for all nonlocal interactions present in the native state. In addition, the contributions of multiple disulfide bonds were considered by summing the partition functions for all the disulfide bonds.

## Computation time

The typical time required to calculate a 2D free energy landscape using the WSME-L model was 13 s for src SH3 domain (60 residues) on a standard desktop computer and 4,680 s for αTS (268 residues) on a 128 CPU parallel supercomputer at the Supercomputer Center, the Institute for Solid State Physics, the University of Tokyo (ISSP). The typical time required to calculate a 2D free energy landscape with the WSME-L(SS$_{intact}$) model was 74 s for BPTI (58 residues, three disulfide bonds) on a standard desktop computer and 3,180 s for RNase A (124 residues, four disulfide bonds) on the 128 CPU parallel supercomputer at ISSP. The calculation of the degree of residue-specific structure formation for theoretical Φ-value analysis requires the above computation time multiplied by the number of residues.

## Kinetic analysis for hen egg-white lysozyme

The microscopic rate constant of the reaction from state X to Y was given by $k_{XY} = A(n_{\ddagger}) \exp(-\Delta F^{\ddagger}_{XY}/k_B T)$, where $\Delta F^{\ddagger}_{XY}$ is the free energy difference between state X and the transition state and is obtained from the 2D free energy landscape. $A(n_{\ddagger})$ is a pre-exponential factor for crossing the transition state at order parameter $n_{\ddagger}$. This value was calculated as $A(n) = A_0/(1 + 5Q(n))$ considering internal friction[23,69]. $Q$ is the degree of contact formation during the folding process. Since this number is approximately proportional to the square of the number of residues, we used $Q = n^2$ ($0 \le n \le 1, 0 \le Q \le 1$) for simplicity. $A_0$ was determined to simulate the experimentally observed folding rates at 293 K[52]. The time-dependent changes in the concentrations of U, I$_1$, I$_2$, and N were calculated as described below.

The kinetic analysis was performed according to the following scheme:

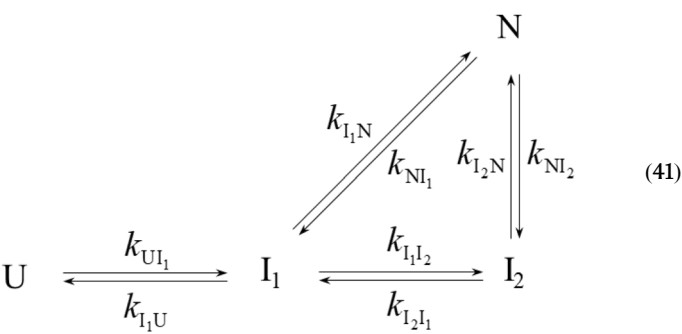

which gives the following matrix equation:

$$\frac{d}{dt}\mathbf{C}(t) = -\mathbf{M}\mathbf{C}(t) \tag{42}$$

where $\mathbf{c} = \begin{pmatrix} U \\ I_1 \\ I_2 \\ N \end{pmatrix}$ and $\mathbf{M} = -\begin{pmatrix} -k_{UI_1} & k_{I_1U} & 0 & 0 \\ k_{UI_1} & -k_{I_1U} - k_{I_1I_2} - k_{I_1N} & k_{I_2I_1} & k_{NI_1} \\ 0 & k_{I_1I_2} & -k_{I_2I_1} - k_{I_2N} & k_{NI_2} \\ 0 & k_{I_1N} & k_{I_2N} & -k_{NI_1} - k_{NI_2} \end{pmatrix}$.

Under native conditions, where the free energy barriers between N and TS$_3$ and between N and TS$_2$ are large, both $k_{I_1N} \gg k_{NI_1}$ and $k_{I_2N} \gg k_{NI_2}$ are satisfied. Then, we can neglect the unfolding of the native

state, as follows:

$$\tag{43}$$

In this scheme, the concentrations of the native states formed through Pathways 1 and 2 are denoted as N$_{p1}$ and N$_{p2}$, respectively. Here, vector **C** and the matrix **M** in Eq. (42) are written as follows:

$$\mathbf{C} = \begin{pmatrix} U \\ I_1 \\ I_2 \\ N_{p1} \\ N_{p2} \end{pmatrix} \text{ and } \mathbf{M} = -\begin{pmatrix} -k_{UI_1} & k_{I_1U} & 0 & 0 & 0 \\ k_{UI_1} & -k_{I_1U} - k_{I_1I_2} - k_{I_1N_{p2}} & k_{I_2I_1} & 0 & 0 \\ 0 & k_{I_1I_2} & -k_{I_2I_1} - k_{I_2N_{p1}} & 0 & 0 \\ 0 & 0 & k_{I_2N_{p1}} & 0 & 0 \\ 0 & k_{I_1N_{p2}} & 0 & 0 & 0 \end{pmatrix} \tag{44}$$

The matrix equation was solved analytically using Mathematica 12.2 (Wolfram), and the time-dependent changes in U, I$_1$, I$_2$, N$_{p1}$, N$_{p2}$, and N (= N$_{p1}$ + N$_{p2}$) were obtained under the following initial conditions: U(0) = 1 and I$_1$(0) = I$_2$(0) = N$_{p1}$(0) = N$_{p2}$(0) = 0.

## Time evolution of domain-specific structure formation in lysozyme

The theoretical Φ-values provided the degree of residue-specific structure formation in the U, I$_1$, I$_2$, and N states of the lysozyme. Using these values, the degrees of α- and β-domain structure formation, Φ$_{th,\alpha}$ and Φ$_{th,\beta}$, were obtained by calculating the average of the Φ$_{th}$-values for the residues involved in the α- and β-domains, respectively. For comparison with experiments[49], the following residues were used for the α-domain: residues 8, 10, 11, 12, 13, 17, 23, 27, 28, 29, 31, 34, 36, 37, 38, 39, 92, 93, 94, 95, 96, 97, 99, 108, 111, 112, 115, 123, 124, and 125, while the following residues were used for the β-domain: residues 40, 42, 44, 50, 52, 53, 56, 58, 61, 63, 64, 65, 75, 76, 78, 82, 83, and 84. Combined with the kinetic analysis, the time evolution of the domain-specific structure formation for the α- and β-domains, $p_\alpha(t)$ and $p_\beta(t)$, respectively, during folding was obtained as follows:

$$\begin{cases} p_\alpha(t) = \Phi_{th,\alpha}(n_U)U(t) + \Phi_{th,\alpha}(n_{I_1})I_1(t) + \Phi_{th,\alpha}(n_{I_2})I_2(t) + \Phi_{th,\alpha}(n_N)N(t) \\ p_\beta(t) = \Phi_{th,\beta}(n_U)U(t) + \Phi_{th,\beta}(n_{I_1})I_1(t) + \Phi_{th,\beta}(n_{I_2})I_2(t) + \Phi_{th,\beta}(n_N)N(t) \end{cases} \tag{45}$$

where $n_U$, $n_{I_1}$, $n_{I_2}$, and $n_N$ are the order parameters of the U, I$_1$, I$_2$, and N states, respectively.

## Thermodynamics of lysozyme folding

The temperature dependence of the heat capacity $C(T)$ was obtained from the partition function as follows:

$$C(T) = \frac{d}{dT}\left(k_B T^2 \frac{d \ln Z}{dT}\right) \tag{46}$$

For comparison with the experimental data from differential scanning calorimetry[54], 8 kcal/(mol·K) was added as a baseline.

According to previous studies on solvation free energy, the temperature dependence of the contact energy size, $\varepsilon$, of lysozyme was

defined as follows:

$$\varepsilon = \varepsilon(T) = \varepsilon_f + p\left(1 - \frac{T}{T_f}\right) - q\left[\left(1 - \frac{T}{T_f}\right) + \frac{T}{T_f}\ln\frac{T}{T_f}\right] \quad (47)$$

where $T_f$ was 293 K[70]. $\varepsilon_f[=\varepsilon(T_f)]$ was set to 1.783 to satisfy the stability of lysozyme ($\Delta G_{NU} = 16k_B T$) at $T_f$. $p$ and $q$ were set to −0.307 and 11.3 to satisfy $\Delta G_{NU}(T_m) = 0$ at a midpoint temperature of thermal unfolding ($T_m = 350$ K) and match the heat capacity observed in experiments[54].

### Simultaneous introduction of multiple disulfide bonds in lysozyme

Generalization of the Hamiltonian with two linkers in the WSME-L(SS$_{intact}$) model yields a Hamiltonian for proteins with multiple linkers. The presence of $L$ linkers at the residue pairs of $(u_1, v_1)$, $(u_2, v_2)$, ..., and $(u_L, v_L)$, in addition to the main chain, provides $(L + 1)$ possible ways to connect the residues $i$ and $j$. Accordingly, the Hamiltonian for multiple linkers is described as follows:

$$H_{ML}(\{m\}) = \sum_{i=1}^{N-1}\sum_{j=i+1}^{N}\varepsilon_{ij}\left[\left(m_{ij} + \sum_{k=1}^{L}m_{ij}^{(u_k,v_k)}\right)/(L+1)\right] \quad (48)$$

The partition function for this Hamiltonian can be solved by repeatedly applying the exact solution for the partition function with a single linker $Z^{(u,v)}(n)$, as many times as the number of linkers. We extracted the appropriate weights from the generating function and introduced additional weights corresponding to the new linkers into the original weights similarly extracted. However, as the number of linkers increased, the formula became more complex, and the computation time increased.

The folding processes of disulfide-intact lysozyme obtained using this method (Supplementary Fig. 44) were almost the same as those obtained using the WSME-L(SS$_{intact}$) model (Fig. 5). As the simultaneous introduction of more than three disulfide bonds complicates the calculations, the WSME-L(SS$_{intact}$) model is suitable for predicting the folding of disulfide-intact proteins.

### Reporting summary

Further information on research design is available in the Nature Portfolio Reporting Summary linked to this article.

## Data availability

The protein structures used in this study are available in PDB under accession codes: 2jwt (En-HD), 4jz4 (src SH3), 1u06 (α-spectrin SH3), 1csp (CspB), 7a1h (CI2), 1aye (ADA2h), 1bzp (apoMb), 1a2p (barnase), 7vsc (RNase HI), 5uih (DHFR), 1jul (IGPS), 1iee (lysozyme), 6etl (RNase A), and 5pti (BPTI) and in AlphaFold Protein Structure Database under accession code: P0A877 (αTS). The computational model structure of apoMb generated in this study is provided in Supplementary Data 1. All data generated or analyzed during this study are included in this article and its Supplementary Information files. Source data are provided with this paper.

## Code availability

The custom codes to generate folding free energy landscapes using the WSME-L, WSME-L(SS), and WSME-L(SS$_{intact}$) models are available at https://github.com/ut-arailab/WSME-L_model, which is archived in Zenodo with the identifier https://doi.org/10.5281/zenodo.8280372[71].

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

## Acknowledgements

This work was supported by JSPS KAKENHI Grant Numbers JP16H02217, JP19H02521, JP21K18841, and JP23H04545 (M.A.), Kayamori Foundation of Informational Science Advancement (M.A.), and a Grant-in-Aid for JSPS Fellows Grant Number JP20J11762 (K.O.). We thank Masaki Sasai and Tomoki P. Terada for valuable discussions and comments. The computations in this work were partially performed using the facilities of the Supercomputer Center, the Institute for Solid State Physics, the University of Tokyo.

## Author contributions

K.O. and M.A. conceived and designed the study, K.O. performed the theoretical calculations, and K.O. and M.A. wrote the paper.

## Competing interests

The authors declare no competing interests.
