## [Peer Review File · Nature Communications]

REVIEWER COMMENTS

Reviewer #1 (Remarks to the Author):

The authors of this work extend the original WSME model, a successful statistical mechanical model of protein folding, to study multi-domain proteins. The work presented is a bit confusing in the way the approach is discussed. Moreover, sufficient effort is not put in to explain how the model is different from the original version applicable to multi-domain proteins with non-contiguous elements developed by Inanami et al (PNAS, 2014). The advantages afforded by the model are minimal and is only a minor step compared to the large number of proteins already characterized by the WSME model. My major comments below:

1) How different is the current approach than that pioneered by Inanami et al (PNAS, 2014)? In the 2014 paper, the authors develop a similar approach to study multi-domain proteins and use DHFR as a model system. It is not clear to me from the manuscript the precise points of difference between the two works. In fact, Figure 2 of the manuscript is eerily similar to Figure 2 of Inanami et al. Also, do the results match between the two works when applied to DHFR?

2) The WSME model is primarily phenomenological in its approach. Therefore, experimental data need to be reproduced before arriving at conclusions. In this work, the authors do not reproduce any experimental works (thermograms, unfolding curves, kinetics, phi-values) but claim agreement at a qualitative level, which is not sufficient, esp. when considering the 'impact' of Nature Communications. A more detailed study needs to be done - are the unfolding curves predicted well? Is there a disagreement? If so, why? The authors do mention specific parameters where they exclude the nearest neighbours in their calculation of contact map. But why and how were these conditions arrived at? Does, for example, the model reproduce the sharp DSC thermogram presented in <https://doi.org/10.1073/pnas.0709881105>?

3) Employing the stability value at 293 K need not necessarily reproduce the melting temperature. Since the model is rapid, it should be possible to predict the melting curves. If there is not an agreement, the model parameters can be tuned to reproduce the experimental data.

4) I am quite surprised that the authors chose to use lysozyme as the only model system. There are plenty of works available in the current literature on multi-domain proteins (including DHFR employed by Inanami et al). Given the simplicity and the rapid approach - to my knowledge, the partition function can be calculated in a matter of seconds for a given temperature - involved in calculating the partition function, the authors should be able to study and demonstrate agreement with many proteins to test the applicability of the model. As of now, the results are too preliminary to be considered useful.

5) Can the authors plot the phi-values as a function of residue number and demonstrate agreement with the predictions? (Is there a phi-value analysis available for lysozyme?) Or hydrogen-exchange protection factors to highlight the agreements?

6) A prior work by Thirumalai and coworkers has shown that the formation of disulphide bonds are entirely determined by the final three dimensional structure and the associated interactions (<https://doi.org/10.1073/pnas.1503909112>). In other words, it is the folding that drives the disulphide bond formation and not vice versa. In the current work, given that the model pre-assumes a specific interaction topology (through the multiple disulphide bond induced loop closure), the assumed ensemble of states will have partially structured states that have the disulphide bridge formed. In other words, I expect the model to pre-assume the formation of disulphide bonds in the unfolded ensemble which will not be correct given the current evidence from literature.

7) The folding barriers - for example, in Fig. 5b - are quite small (a few kBT) compared to the slow folding of lysozyme which is of the order of milliseconds or slower. In other words, one would expect the barrier to be significantly larger which is not evident from the figures. Can the authors explain this apparent discrepancy?

Reviewer #2 (Remarks to the Author):

Three-dimensional structure based statistical mechanical models for calculating the energy landscape of single domain proteins have revealed insights into protein folding owing, in part, to their simplicity. One of the limitations of these models thus far, however, is that they are not applicable to multi-domain proteins containing native interactions between sequence-distant residues. Ooka and Arai address this limitation by introducing linkers in the stat mech model that allow sequence distant residues to interact without the requirement that all intervening residues be in a native conformation. The idea had been suggested over a decade ago, but this work is the first successful implementation.

The calculation of the energy landscape of lysozyme is used to demonstrate the model, referred to as the WSME-L model. Lysozyme is an excellent test case given that there are four disulfide bonds and a wealth of experimental data for comparison. The WSME-L model is successful in capturing the salient features of the energy landscape determined from kinetic studies. The single disulfide bond studies further reinforce the role of the linker and the consistency with experimental findings. Overall, the agreement between experiment and calculation is very good. WSME-L is an important and welcome extension of the WSME model and will be a valuable tool for those in the protein sciences interested in

gaining insights into the folding mechanism and higher energy partially folded states of a protein of interest with a known structure.

My only reservation about this work is that it was demonstrated using only one protein, lysozyme. Admittedly, lysozyme is an excellent choice, but I was surprised that the authors did not include multi-domain proteins without disulfides to show the improvement provided by WSME-L vs WSME (e.g., some of the examples in reference 14 and some circular permutants come to mind). The reasons for this weren't clear to me and it's possible I might have simply missed an obvious consideration. My understanding is that one of the advantages of models such as WSME (and, presumably, WSME-L) is that they can be performed relatively quickly. Does the need to determine the AMBER-derived contact energy slow things down? Additional details on the limitations of performing the WSME-L calculation on other proteins would be helpful.

Very minor point:

In their discussion of the comparison between calculation and experiment the authors note a difference of 60% vs 45% for the maximum population of the I2 intermediate. This seems very minor and quite impressive given the simplicity of the model.

Response to Reviewers

Reviewer #1

The authors of this work extend the original WSME model, a successful statistical mechanical model of protein folding, to study multi-domain proteins. The work presented is a bit confusing in the way the approach is discussed. Moreover, sufficient effort is not put in to explain how the model is different from the original version applicable to multi-domain proteins with non-contiguous elements developed by Inanami et al (PNAS, 2014). The advantages afforded by the model are minimal and is only a minor step compared to the large number of proteins already characterized by the WSME model. My major comments below:

1) How different is the current approach than that pioneered by Inanami et al (PNAS, 2014)? In the 2014 paper, the authors develop a similar approach to study multi-domain proteins and use DHFR as a model system. It is not clear to me from the manuscript the precise points of difference between the two works. In fact, Figure 2 of the manuscript is eerily similar to Figure 2 of Inanami et al. Also, do the results match between the two works when applied to DHFR?

Thank you very much for your comments. Inanami *et al.* previously introduced a virtual linker between the N- and C-termini of a protein, dihydrofolate reductase (DHFR), to account for non-local interactions between the N- and C-terminal regions (ref. 23 of the revised manuscript). The partition function for their “extended WSME model” was as follows:

$$Z_{\text{eWSME}} = Z + (Z^{(N,C)} - Z) \exp(S^{(N,C)} / k_B),$$

where Z is the partition function of the original WSME model, $Z^{(N,C)}$ is the partition function for the protein in which the N- and C-termini are connected by a virtual linker, $S^{(N,C)}$ is the entropic cost of bringing the N- and C-termini to the positions in the native conformations, and k_B is the Boltzmann constant. The second term includes $Z^{(N,C)} - Z$ to exclude the contribution of protein conformations without the virtual linker from $Z^{(N,C)}$. Although this model was applicable to DHFR, in which the N- and C-termini are in close proximity, it was not applicable to the proteins with distant N- and C-termini. In addition, the model could not account for all of the non-local interactions present in the protein. To address these issues, we have constructed a WSME-L model that allows the introduction of a virtual linker between any amino acid residues, allowing the introduction of non-local interactions at any site within the protein. In the revised manuscript, we have clearly described this point in page 5, lines 106–109 as follows:

“To consider the nonlocal interactions between the N- and C-termini of a protein, Inanami et al. introduced a virtual linker at both termini²³. Inspired by this, we developed a method to introduce a virtual linker between arbitrary residues u and v ($u < v$) and established a new model (WSME-L) that can consider all nonlocal interactions in a protein molecule.”

Lysozyme is one of the best studied proteins in terms of protein folding. Its folding mechanisms have been studied in detail in the presence of disulfide bonds throughout the folding studies. Therefore, in the original manuscript, to account for the folding mechanisms of the disulfide-intact lysozyme, we constructed the WSME-L model by introducing four covalent linkers at the positions of its four non-local disulfide bonds. The model have successfully reproduced the experimental results of lysozyme folding. However, as Reviewer 1 has pointed out in comment #6, disulfide bonds should be formed during folding reactions *in vivo* and are not preformed before folding. Therefore, it is necessary to construct a model that can account for the folding reaction involving oxidative disulfide bond formation. In addition, as pointed out by Reviewer 1 in comment #4, it is necessary to develop a model that can be applied to various proteins in addition to lysozyme. If we can construct a model that can account for all non-local interactions present in a protein in addition to disulfide bonds, such a theoretical model will be generally applicable to many proteins with or without disulfide bonds.

Therefore, in the revised manuscript, we greatly extended the WSME-L model described in the original manuscript and constructed a new model that can account for the contribution of all non-local interactions in a protein by considering all native contacts using virtual linkers. We have renamed the new model the WSME-L model. In this model, we used a Hamiltonian $H^{(u,v)}$ that can account for a virtual linker between arbitrary residues u and v as follows:

$$H^{(u,v)}(\{m\}) = \sum_{i=1}^{N-1} \sum_{j=i+1}^N \varepsilon_{i,j} \left[\frac{m_{i,j} + m_{i,j}^{(u,v)}}{2} \right].$$

Its partition function $Z^{(u,v)}$ is described as follows:

$$Z^{(u,v)} = \text{Tr}_n \exp \left[- \left(H^{(u,v)}(\{m\}) - T \sum_{i=1}^N S_i m_i \right) / k_B T \right].$$

These equations are the same as those described in our original manuscript. In the revised WSME-L model, to account for all native contacts between residues u and v using virtual linkers, we summed $Z^{(u,v)}$ for all native contacts and defined the partition function of the new WSME-L model as follows:

$$Z_L = Z + \sum_{(u,v):\text{All contacts}} \left(Z^{(u,v)} - Z \right) \exp \left(S^{(u,v)} / k_B \right),$$

where Z is the partition function of the original WSME model, $Z^{(u,v)}$ is the partition function for the protein in which residues u and v are connected by a virtual linker, and $S^{(u,v)}$ is the entropic cost of bringing residues u and v to the positions in the native conformations. The summation in the second term is computed for all native contacts between residues u and v . This WSME-L model corresponds to a generalization of the virtual linker model by Inanami *et al.* and can consider all native contacts present in the protein. Moreover, we have derived an exact analytical solution for the partition function of the WSME-L model using the transfer matrix method. Such an analytical solution was not reported in the paper by Inanami *et al.* Our computational method allows a significant reduction in computation time: ~ 10 s for the calculation of a rigorous two-dimensional free energy landscape of a small protein using the WSME-L model. This paves the way for the application of the WSME-L model to a wide variety of proteins.

First, we applied this model to predict the folding mechanisms of six small single-domain proteins: engrailed homeodomain (En-HD), src SH3 domain, α -spectrin SH3 domain, cold shock protein B (CspB), chymotrypsin inhibitor 2 (CI2), and activation domain of human procarboxypeptidase A2 (ADA2h). The results showed that our WSME-L model improved the prediction accuracy of the Φ -values, which indicate the degree of structure formation of the transition state, compared to the original WSME models. We also applied the WSME-L model to predict the folding of six large multidomain proteins: apomyoglobin (apoMb), barnase, ribonuclease HI (RNase HI), DHFR, α -subunit of tryptophan synthase (α TS), and indole-3-glycerol phosphate synthase (IGPS). α TS (268 residues) and IGPS (223 residues) are among the largest proteins whose folding has been studied in detail by experiment. Surprisingly, our model accurately predicted the experimentally observed folding behavior of these proteins. Moreover, the model was also able to predict different folding mechanisms for two proteins with similar structures as observed in the experiments.

For DHFR, the extended WSME model by Inanami *et al.* with a virtual linker between the N- and C-termini partially explained the experimental results. However, the predicted intermediates were unstable, and multiple folding pathways were not predicted. In contrast, our WSME-L model reproduced the accumulation of stable intermediates and the presence of multiple folding pathways, improving the prediction accuracy.

We then slightly modified the WSME-L model to construct a model that can predict folding reactions involving oxidative disulfide bond formation (named the WSME-L(SS) model). This model is basically the same as the WSME-L model described above, except that the large stabilization energy is added only when a disulfide bond is formed during the folding reaction. Disulfide bonds are not formed prior to folding, and whether or not a disulfide bond is formed depends on whether the surrounding region is energetically favorable for folding. Thus, the WSME-L(SS) model can explain oxidative folding with disulfide bond formation. We found that this model can successfully reproduce the experimental results of oxidative folding of three representative proteins: lysozyme, ribonuclease A (RNase A), and bovine pancreatic trypsin inhibitor (BPTI).

We further modified this model to construct a WSME-L(SS_{intact}) model that can account for the folding of proteins with one or more disulfide bonds intact. In this model, the partition function is calculated similarly to the WSME-L model, but under the conditions that a single disulfide bond between residues u_2 and v_2 is preformed by a covalent linker (i.e., a virtual linker without considering the entropic cost of linker formation). If there are multiple disulfide bonds, the partition function $Z_L^{SS \text{ intact}}$ for the protein was obtained by calculating the partition functions of the WSME-L model in which only one of the disulfides is preformed and then summing them. The model was successfully applied to two proteins: lysozyme and RNase A. The predicted intermediate structures were consistent with experimental results at the amino acid residue level. Moreover, the predicted thermodynamics and kinetics of lysozyme folding were quantitatively consistent with experimental results (see also responses to comments #2 and #5 of Reviewer 1).

In our original manuscript, we calculated the partition function for the simultaneous presence of four disulfide bonds in lysozyme and did not introduce virtual linkers for non-covalent native contacts. This model gave almost the same free energy landscape as the WSME-L(SS_{intact}) model, but the simultaneous introduction of multiple disulfide bonds required a complicated formulation and long computation time. In contrast, the WSME-L(SS_{intact}) model considers the

contribution of each disulfide bond to the partition function in an additive manner, which makes the calculation more efficient.

In summary, we have significantly improved the WSME-L model in the revised manuscript to predict folding mechanisms without the limitations of protein size and shape. Furthermore, slight modifications of the model allowed us to develop the WSME-L(SS) model for predicting oxidative folding and the WSME-L(SS_{intact}) model for predicting folding of disulfide-intact proteins. Using these models, we succeeded in accurately predicting 17 different protein folding reactions. Thus, we believe that our model is an excellent model for predicting folding mechanisms regardless of protein size, shape, and the presence or absence of disulfide bonds, and represents a major advance over the original WSME models and the extended WSME model by Inanami *et al.* The above revisions have been made in the revised manuscript as follows:

- lines 69–85 in Introduction
- lines 126–349 in Results
- lines 357–400 in Discussion
- lines 418–506, 573–676, and 725–778 in Methods
- Figures 1–5
- Supplementary Tables 1–2
- Supplementary Figures 1–28 and 37–40

2) The WSME model is primarily phenomenological in its approach. Therefore, experimental data need to be reproduced before arriving at conclusions. In this work, the authors do not reproduce any experimental works (thermograms, unfolding curves, kinetics, phi-values) but claim agreement at a qualitative level, which is not sufficient, esp. when considering the 'impact' of Nature Communications. A more detailed study needs to be done - are the unfolding curves predicted well? Is there a disagreement? If so, why? The authors do mention specific parameters where they exclude the nearest neighbours in their calculation of contact map. But why and how were these conditions arrived at? Does, for example, the model reproduce the sharp DSC thermogram presented in <https://doi.org/10.1073/pnas.0709881105>?

Thank you for your comment. As mentioned in the response to comment #1, in the revised manuscript, we have constructed three different WSME-L models to predict the folding reactions of various proteins. Using the WSME-L(SS_{intact}) model, we were able to quantitatively reproduce the thermal denaturation curves of lysozyme measured by DSC reported in the above literature. The results are shown in Fig. 5h of the revised manuscript. In addition, the time evolution of the α - and β -domain structure formation of lysozyme, calculated by combining the kinetic and theoretical Φ -value analyses, reproduced well the experimental results obtained by pulsed-hydrogen exchange NMR measurements. The results are shown in Fig. 5g of the revised manuscript. Thus, the WSME-L(SS_{intact}) model can quantitatively explain the thermodynamics and kinetics of lysozyme folding.

In addition, the predictions of the WSME-L model for the six small proteins were in excellent agreement with the experimental results for folding rate and native-state stability, with correlation coefficients of 0.919 and 0.995, respectively (Fig. 2g, h of the revised manuscript). Moreover, the Φ -values at the transition state predicted by the WSME-L model were more

consistent with the experimental results than the original WSME models. For example, the Φ -values of the src SH3 domain predicted by the original WSME model did not agree with experimental values (correlation coefficient $r = 0.084$), whereas the Φ -values predicted by the WSME-L model agreed very well with experimental values ($r = 0.870$) (Fig. 2f of the revised manuscript). Furthermore, the WSME-L model reproduced the Φ -values for the folding of large proteins, barnase and RNase HI, with higher accuracy than the original WSME model (Supplementary Figs. 11, 13 of the revised manuscript). Thus, the WSME-L models can quantitatively reproduce folding reactions of small to large proteins.

In our calculations we excluded the contribution of native contacts between neighboring residues in an amino acid sequence. The reason for this is that residues close in a sequence are always in close proximity during the folding reaction, so their contact energies between the unfolded and native states should not be different. To our knowledge, most previous studies using the WSME model have used the same treatment. Therefore, we believe that this treatment for neighboring contacts is reasonable.

3) Employing the stability value at 293 K need not necessarily reproduce the melting temperature. Since the model is rapid, it should be possible to predict the melting curves. If there is not an agreement, the model parameters can be tuned to reproduce the experimental data.

As noted in our response to comment #2, we have reproduced the thermal denaturation curve of lysozyme measured by DSC using the WSME-L(SS_{intact}) model. In this calculation, the temperature dependence was rigorously calculated by considering a hydration free energy term in the contact energy of lysozyme, according to the previous study (Naganathan (2012) *J Chem Theory Comput* 8, 4646-4656). By setting the parameters as described in Methods section of the revised manuscript (lines 741–757), we were able to reproduce the melting temperature and the thermal denaturation curve as well as the stability at 293 K. The results are shown in Fig. 5h of the revised manuscript.

4) I am quite surprised that the authors chose to use lysozyme as the only model system. There are plenty of works available in the current literature on multi-domain proteins (including DHFR employed by Inanami et al). Given the simplicity and the rapid approach - to my knowledge, the partition function can be calculated in a matter of seconds for a given temperature - involved in calculating the partition function, the authors should be able to study and demonstrate agreement with many proteins to test the applicability of the model. As of now, the results are too preliminary to be considered useful.

Thank you for your comments. In accordance with the Reviewer's comments, we have modified the WSME-L model for application to many proteins, as described above. We then used the WSME-L models to predict 17 protein folding reactions: folding of six small single-domain proteins with different structures, folding of six large multidomain proteins with different structures,

folding involving oxidative disulfide bond formation for three proteins, and folding of two proteins with intact disulfide bonds. These predictions are consistent with experiment and suggest that the WSME-L models are useful and generally applicable to a wide variety of proteins, regardless of size and shape. We have included these results in the revised manuscript, as noted in our response to Reviewer 1's comment #1.

5) Can the authors plot the phi-values as a function of residue number and demonstrate agreement with the predictions? (Is there a phi-value analysis available for lysozyme?) Or hydrogen-exchange protection factors to highlight the agreements?

Since the experimental Φ -value analysis has not been reported for lysozyme, we attempted to reproduce the experimental results of the pulsed-hydrogen-exchange NMR measurements. By combining theoretical Φ -value analysis with kinetic analysis, we were able to quantitatively reproduce the time evolution of the α - and β -domain folding measured by proton occupancy. The results are shown in Fig. 5g of the revised manuscript. In addition, we calculated theoretical Φ -values using the WSME-L model for six small proteins and two large proteins for which Φ -values in the transition state have been measured experimentally at the amino acid residue level. The predicted values were well correlated with the experimental results. These results indicate that our WSME-L model can predict folding reactions with high accuracy at the amino acid residue level. We have described these results in lines 167–177 and 199–204, Fig. 2, and Supplementary Figs. 1–6, 11, and 13 of the revised manuscript.

6) A prior work by Thirumalai and coworkers has shown that the formation of disulphide bonds are entirely determined by the final three dimensional structure and the associated interactions (<https://doi.org/10.1073/pnas.1503909112>). In other words, it is the folding that drives the disulphide bond formation and not vice versa. In the current work, given that the model pre-assumes a specific interaction topology (through the multiple disulphide bond induced loop closure), the assumed ensemble of states will have partially structured states that have the disulphide bridge formed. In other words, I expect the model to pre-assume the formation of disulphide bonds in the unfolded ensemble which will not be correct given the current evidence from literature.

Thank you for the comment. We agree with the Reviewer that in oxidative folding reactions involving disulfide bond formation, the folding reaction induces the formation of disulfide bonds, as shown by Thirumalai *et al.* It is also known that residual structures can exist in the unfolded state when disulfide bonds are present. Therefore, in the revised manuscript, we have constructed a WSME-L(SS) model that can predict the oxidative folding reaction of lysozyme, RNase A, and BPTI from the fully unfolded state in which all disulfide bonds are reduced. These proteins are among those for which the oxidative folding process has been best studied. The predictions of the WSME-L(SS) model were in good agreement with the experimental results. A detailed comparison of the experimental results with the predictions can be found in the revised

manuscript (lines 232–288).

Because of the low temporal resolution of the above oxidative folding experiments, Tanford, Dobson, Kiefhaber, and many others have measured the refolding reactions of proteins with intact disulfide bonds in detail with millisecond temporal resolution. We have reproduced these experimental results using the WSME-L(SS_{intact}) model. In the original manuscript, the prediction results for lysozyme were described in detail, but in the revised manuscript, the description of lysozyme has been greatly shortened because we have predicted the folding of many other proteins.

The WSME-L(SS) model is essentially the same as the WSME-L model without disulfide bond formation, where all contacts present in the native state are considered by a virtual linker. In the WSME-L(SS) model, only if a disulfide bond is formed during the folding reaction, the large stabilization energy by disulfide formation is added. Therefore, the disulfide bond is not formed in advance, and whether or not a disulfide bond can be formed depends on whether the surrounding region is energetically favorable for folding. This is consistent with the idea of Thirumalai *et al.* that folding promotes disulfide bond formation.

7) The folding barriers - for example, in Fig. 5b - are quite small (a few $k_B T$) compared to the slow folding of lysozyme which is of the order of milliseconds or slower. In other words, one would expect the barrier to be significantly larger which is not evident from the figures. Can the authors explain this apparent discrepancy?

As Reviewer 1 pointed out, the free energy barrier from I₁ to N in Pathway 2 of lysozyme folding was about 5 $k_B T$. However, it has been reported that internal friction slows down the folding rate of proteins as the folding reaction proceeds. Since the free energy landscape calculated from the WSME-L model does not account for internal friction, more time may be required to surpass this barrier. Indeed, Inanami *et al.* (ref. 23) previously showed with Monte Carlo simulations considering internal friction that it takes tens of milliseconds to cross the barrier of a few $k_B T$ in the later stages of the folding reaction. We therefore calculated the folding rate by introducing internal friction as in Inanami *et al.* The results showed that the folding rate is consistent with the experiment for a reasonable range of parameters. Furthermore, the time evolution of the structure formation of the α - and β -domains of lysozyme, calculated in conjunction with theoretical Φ -value analysis, reproduced well the kinetic experimental results from the pulsed-hydrogen exchange NMR measurements. These results are described in lines 316–324 and 678–739 and Fig. 5g of the revised manuscript.

Reviewer 2

Three-dimensional structure based statistical mechanical models for calculating the energy landscape of single domain proteins have revealed insights into protein folding owing, in part, to their simplicity. One of the limitations of these models thus far, however, is that they are not applicable to multi-domain proteins containing native interactions between sequence-distant residues. Ooka and Arai address this limitation by introducing linkers in the stat mech model that allow sequence distant residues to interact without the requirement that all intervening residues be in a native conformation. The idea had been suggested over a decade ago, but this work is the first successful implementation.

Thank you very much for understanding the significance of our research.

My only reservation about this work is that it was demonstrated using only one protein, lysozyme. Admittedly, lysozyme is an excellent choice, but I was surprised that the authors did not include multi-domain proteins without disulfides to show the improvement provided by WSME-L vs WSME (e.g., some of the examples in reference 14 and some circular permutants come to mind). The reasons for this weren't clear to me and it's possible I might have simply missed an obvious consideration. My understanding is that one of the advantages of models such as WSME (and, presumably, WSME-L) is that they can be performed relatively quickly. Does the need to determine the AMBER-derived contact energy slow things down? Additional details on the limitations of performing the WSME-L calculation on other proteins would be helpful.

Thank you for your comment. In the original manuscript, we focused on the folding of disulfide-intact lysozyme because folding prediction of disulfide-intact proteins was one of the most difficult problems in the extension of the WSME model. However, as Reviewers 1 and 2 pointed out, it is important to build a model that can be applied to many proteins. Therefore, we have substantially improved the WSME-L model described in the original manuscript to construct a model that is generally applicable to all proteins, and we have renamed this new model the WSME-L model.

First, we applied this model to predict the folding mechanisms of six small single-domain proteins: engrailed homeodomain (En-HD), src SH3 domain, α -spectrin SH3 domain, cold shock protein B (CspB), chymotrypsin inhibitor 2 (CI2), and activation domain of human procarboxypeptidase A2 (ADA2h). The results showed that our WSME-L model improved the prediction accuracy of the Φ -values, which indicate the degree of structure formation of the transition state, compared to the original WSME models. We also applied the WSME-L model to predict the folding of six large multidomain proteins: apomyoglobin (apoMb), barnase, ribonuclease HI (RNase HI), DHFR, α -subunit of tryptophan synthase (α TS), and indole-3-glycerol phosphate synthase (IGPS). α TS (268 residues) and IGPS (223 residues) are among the largest proteins whose folding has been studied in detail by experiment. Surprisingly, our model accurately predicted the experimentally observed folding behavior of these proteins. Moreover, the model was also able to predict different folding mechanisms for two proteins with similar structures as observed in the experiments.

We then slightly modified the WSME-L model to construct a model that can predict

folding reactions involving oxidative disulfide bond formation (named the WSME-L(SS) model). This model is basically the same as the WSME-L model described above, except that the large stabilization energy is added only when a disulfide bond is formed during the folding reaction. We found that the WSME-L(SS) model can successfully reproduce the experimental results of oxidative folding of three representative proteins: lysozyme, ribonuclease A (RNase A), and bovine pancreatic trypsin inhibitor (BPTI).

We further modified this model to construct a WSME-L(SS_{intact}) model that can account for the folding of disulfide-intact proteins. The model was successfully applied to two proteins: lysozyme and RNase A. The predicted intermediate structures were consistent with experimental results at the amino acid residue level. Moreover, the predicted thermodynamics and kinetics of lysozyme folding were quantitatively consistent with experimental results.

In summary, we have significantly improved the WSME-L model in the revised manuscript to predict folding mechanisms without the limitations of protein size and shape. Furthermore, slight modifications of the model allowed us to develop the WSME-L(SS) model for predicting oxidative folding and the WSME-L(SS_{intact}) model for predicting folding of disulfide-intact proteins. Using these models, we succeeded in accurately predicting 17 different protein folding reactions. Thus, we believe that our model is an excellent model for predicting folding mechanisms regardless of protein size, shape, and the presence or absence of disulfide bonds, and represents a major advance over the original WSME models and the extended WSME model by Inanami *et al.* (ref. 23). The above revisions have been made in the revised manuscript, as described in the response to comment #1 of Reviewer 1.

Detailed comparisons of predicted and experimental data reveal slight differences between them due to non-native interactions and Pro isomerization during folding. Experiments have shown that in disulfide-intact lysozyme folding, non-native interactions stabilize the α -domain intermediate (I_2) and retard the subsequent folding process. The WSME-L prediction underestimated the maximum population of I_2 (~45%) compared to the experimental results (~60%), but this discrepancy can be resolved by accounting for the stabilization effect of non-native interactions by 0.44 kcal/mol. Experiments have also shown that in α TS and IGPS folding, off-pathway intermediates are formed by non-native interactions and that *cis* Pro residues in the native state of α TS, IGPS, and RNase A must isomerize from *trans* to *cis* conformations during folding if they are in *trans* conformations in the unfolded state, resulting in the appearance of additional parallel folding pathways. However, the WSME-L models predicted only the folding pathways involving on-pathway intermediates with native-like Pro conformations because they only consider native contacts. Similarly, the WSME-L(SS) model predicted that the 40–95 disulfide bond is formed together with the 26–84 disulfide bond in the final step of oxidative RNase A folding, whereas experiments have shown that the final step of the folding involves only the formation of the 40–95 disulfide bond. This may be because the slow *trans*-to-*cis* isomerization of Pro93 near the 40–95 disulfide bond, which is not considered in the WSME-L(SS) model, delays the formation of this disulfide. Thus, the WSME-L models are useful for capturing the general features of the folding mechanisms of ideal proteins, and slight discrepancies between the predicted and experimental data can provide clues to understanding the role of non-native interactions/conformations in protein folding. Future efforts to incorporate non-native interactions and Pro isomerization into the models would allow for even greater accuracy in predicting protein folding mechanisms. We described this discussion in lines 377–400 of the revised manuscript.

Very minor point:

In their discussion of the comparison between calculation and experiment the authors note a difference of 60% vs 45% for the maximum population of the I₂ intermediate. This seems very minor and quite impressive given the simplicity of the model.

In the WSME-L model, contact energy calculations are performed using the AMBER force field, which allows us to predict folding behavior close to that of real proteins. In fact, we could not obtain a free energy landscape that is consistent with experiment using the WSME-L model when contact energies were not calculated using the AMBER force field. This result is shown in Supplementary Fig. 42 and lines 444–452 of the revised manuscript

The close agreement between the predicted and experimental values of the I₂ population during lysozyme folding (45% vs. 60%) is also a surprise to us and confirms the high prediction accuracy of our model.

Finally, in revising the manuscript, we have developed the WSME-L models that can generally predict the folding mechanisms of a wide range of proteins. Therefore, we would like to change the title of this paper to "Simple statistical mechanical models for predicting protein folding mechanisms".

We believe that the revisions to the manuscript address all of the Reviewers' comments and that the manuscript is now suitable for publication in *Nature Communications*. We thank the editors and all the reviewers again for their efforts during the review process.

REVIEWERS' COMMENTS

Reviewer #1 (Remarks to the Author):

The authors have thoroughly addressed all of my concerns. They have not only done so, but improved the original model proposed in the previous submission at different levels to be widely applicable to multiple model systems. In addition, they show excellent agreement between model predictions, and different experimental data including thermograms, phi-values, folding pathways and relative stability metrics. Given the large body of work, direct agreement with experimental data and the rapidity of the method, I recommend the manuscript for publication.

I, however, have one minor comment:

Can the authors reconsider the title? Given that there are many models available for predicting protein folding mechanisms, it should be possible to refine the title to make it more specific and attractive, but without making it too long. This is my opinion and I do not mandate it.

Reviewer #2 (Remarks to the Author):

In response to the reviewer comments the authors have made changes and additions that significantly improve the manuscript. Specifically, the manuscript now includes predictions and comparisons with experimental data for a number of proteins of various sizes with varying complexity in their energy landscape. Comparison of the WSME-L predictions with hydrogen exchange data are especially useful. Given space constraints the authors did not touch on predictions of HX on all of the proteins but perhaps this will be included in a future manuscript. For the proteins I'm familiar with the predictions of the WSME-L model and HX data look promising at first glance and capture the salient features of the energy landscape. I applaud the authors for making the effort to include the additional proteins and their positive and proactive response to the reviewer comments. The manuscript is an outstanding contribution to protein science that will be useful in both academic labs and also in biotechnology applications (e.g., disulfides are very important in the manufacturing of some biomolecules).

Side note: In looking through the supplementary materials I noticed that the authors have included their source code so that others can use their model on other proteins. Thank you for sharing.

Point-by-point Response to the Reviewers' Comments

Reviewer #1

The authors have thoroughly addressed all of my concerns. They have not only done so, but improved the original model proposed in the previous submission at different levels to be widely applicable to multiple model systems. In addition, they show excellent agreement between model predictions, and different experimental data including thermograms, phi-values, folding pathways and relative stability metrics. Given the large body of work, direct agreement with experimental data and the rapidity of the method, I recommend the manuscript for publication.

Thank you very much for understanding the significance of our research.

I, however, have one minor comment:

Can the authors reconsider the title? Given that there are many models available for predicting protein folding mechanisms, it should be possible to refine the title to make it more specific and attractive, but without making it too long. This is my opinion and I do not mandate it.

Thank you very much for your comments. As suggested by Reviewer 1, we have changed the title of this paper to “Accurate prediction of protein folding mechanisms by simple structure-based statistical mechanical models”.

Reviewer 2

In response to the reviewer comments the authors have made changes and additions that significantly improve the manuscript. Specifically, the manuscript now includes predictions and comparisons with experimental data for a number of proteins of various sizes with varying complexity in their energy landscape. Comparison of the WSME-L predictions with hydrogen exchange data are especially useful. Given space constraints the authors did not touch on predictions of HX on all of the proteins but perhaps this will be included in a future manuscript. For the proteins I'm familiar with the predictions of the WSME-L model and HX data look promising at first glance and capture the salient features of the energy landscape. I applaud the authors for making the effort to include the additional proteins and their positive and proactive response to the reviewer comments. The manuscript is an outstanding contribution to protein

science that will be useful in both academic labs and also in biotechnology applications (e.g., disulfides are very important in the manufacturing of some biomolecules).

Thank you very much for understanding the importance of our research. As pointed out by Reviewer 2, we would like to include in a future paper a comparison of the experimental data and the WSME-L predictions of hydrogen exchange during folding of proteins other than lysozyme.

Side note: In looking through the supplementary materials I noticed that the authors have included their source code so that others can use their model on other proteins. Thank you for sharing.

The source code is now available at https://github.com/ut-arailab/WSME-L_model, which is archived in Zenodo with the identifier [\[https://doi.org/10.5281/zenodo.8280372\]](https://doi.org/10.5281/zenodo.8280372). This information is described in the "Code availability" section of the revised manuscript. We hope that many researchers in the protein science community will find this program useful.

We believe that the revisions to the manuscript address all of the Reviewers' comments and that the manuscript is now suitable for publication in *Nature Communications*. We thank the editors and all the reviewers again for their efforts during the review process.